# Sequential selection of economic good and action in medial frontal cortex of macaques during value-based decisions

Xiaomo Chen[1]*, Veit Stuphorn[1,2,3]*

[1]Department of Psychological and Brain Sciences, Johns Hopkins University, Baltimore, United States; [2]Department of Neuroscience, Johns Hopkins University School of Medicine, Baltimore, United States; [3]Zanvyl Krieger Mind/Brain Institute, Johns Hopkins University School of Medicine, Baltimore, United States

**Abstract** Value-based decisions could rely either on the selection of desired economic goods or on the selection of the actions that will obtain the goods. We investigated this question by recording from the supplementary eye field (SEF) of monkeys during a gambling task that allowed us to distinguish chosen good from chosen action signals. Analysis of the individual neuron activity, as well as of the population state-space dynamic, showed that SEF encodes first the chosen gamble option (the desired economic good) and only ~100 ms later the saccade that will obtain it (the chosen action). The action selection is likely driven by inhibitory interactions between different SEF neurons. Our results suggest that during value-based decisions, the selection of economic goods precedes and guides the selection of actions. The two selection steps serve different functions and can therefore not compensate for each other, even when information guiding both processes is given simultaneously.

*For correspondence: xiaomo@ stanford.edu (XC); veit@jhu.edu (VS)

Competing interests: The authors declares that no competing interests exist.

## Introduction

Value-based decision-making requires the ability to select the reward option with the highest available value, as well as the appropriate action necessary to obtain the desired option. Currently it is still unclear how the brain compares value signals and uses them to select an action (*Gold and Shadlen, 2007*; *Cisek, 2012*). The goods-based model of decision-making (*Padoa-Schioppa, 2011*) suggests that the brain computes the subjective value of each offer, selects one of these option value signals, and then prepares the appropriate action plan (*Figure 1A*). Support for this model comes from recording studies in orbitofrontal cortex (OFC) during an economic choice task (*Padoa-Schioppa and Assad, 2006*; *Cai and Padoa-Schioppa, 2012*). In contrast, the action-based model of decision making (*Tosoni et al., 2008*; *Cisek and Kalaska, 2010*; *Christopoulos et al., 2015a*) suggests that all potential actions are represented in the brain in parallel and compete with each other (*Figure 1B*). This competition is influenced by a variety of factors including the value of each actions' outcome. According to this model, option value signals should not predict the chosen option, since these signals only serve as input into the decision process, which is determined by the competition among the potential actions. Support for this model comes primarily from recording studies in parietal and premotor cortex (*Platt and Glimcher, 1999*; *Sugrue et al., 2004*; *Shadlen et al., 2008*; *Cisek and Kalaska, 2010*; *Christopoulos et al., 2015b* ).

As there is evidence supporting both theories, it is unlikely that either the goods-based or the action-based model in their pure form are correct. However, the exact role of goods- and action-based selection processes in decision making is not known. The distributed consensus model (*Cisek, 2012*) combines elements of the goods-based and the action-based model (*Figure 1C*). It is

**eLife digest** Much of our decision making seems to involve selecting the best option from among those currently available, and then working out how to attain that particular outcome. However, while this might sound straightforward in principle, exactly how this process is organized within the brain is not entirely clear.

One possibility is that the brain compares all the possible outcomes of a decision with each other before constructing a plan of action to achieve the most desirable of these. This is known as the 'goods-based' model of decision making. However, an alternative possibility is that the brain instead considers all the possible actions that could be performed at any given time. One specific action is then chosen based on a range of factors, including the potential outcomes that might result from each. This is an 'action-based' model of decision making.

Chen and Stuphorn have now distinguished between these possibilities by training two monkeys to perform a gambling task. The animals learned to make eye movements to one of two targets on a screen to earn a reward. The identity of the targets varied between trials, with some associated with larger rewards or a higher likelihood of receiving a reward than others. The location of the targets also changed in different trials, which meant that the choice of 'action' (moving the eyes to the left or right) could be distinguished from the choice of 'goods' (the reward).

By using electrodes to record from a region of the brain called the supplementary eye field, which helps to control eye movements, Chen and Stuphorn showed that the activity of neurons in this region predicted the monkeys' decision-making behavior. Crucially, it did so in two stages: neurons first encoded the reward chosen by the monkey, before subsequently encoding the action that the monkey selected to obtain that outcome.

These data argue against an action-based model of decision making because outcomes are encoded before actions. However, they also argue against a purely goods-based model. This is because all possible actions are encoded by the brain (including those that are subsequently rejected), with the highest levels of activity seen for the action that is ultimately selected. The data instead support a new model of decision making, in which outcomes and actions are selected sequentially via two independent brain circuits.

characterized by strong reciprocal interactions between the goods and the action representation levels that allow the action selection to influence the simultaneous ongoing value selection and vice versa. This model predicts therefore that the selection of the chosen good and action are closely integrated and proceed in parallel.

Here, we test these different models by recording neuronal activity in the supplementary eye field (SEF). Previous research indicates that neurons in the SEF participate in the use of value signals to select eye movements (*So and Stuphorn, 2010*). Its anatomical connections make the SEF ideally suited for this role. It receives input from areas that represent option value, such as OFC, ACC, and the amygdala (*Huerta and Kaas, 1990*; *Matsumoto et al., 2003*; *Ghashghaei et al., 2007*), and projects to oculomotor areas, such as frontal eye field (FEF) and superior colliculus (*Huerta and Kaas, 1990*).

We designed an oculomotor gamble task, in which the monkey had to choose between two gamble options indicated by visual cues. The monkeys indicated their choice by making a saccade to the cue indicating the desired gamble option. Across different trials, the visual cues were presented in different locations and required saccades in different directions to be chosen. This allowed us to distinguish the selection of gamble options or economic goods from the selection of actions. We found that the activity of SEF neurons predicted the monkey's choice. Importantly, this decision process unfolded sequentially. First, the chosen gamble option was selected and only then the chosen action. The saccade selection process seemed to be driven by competition between directionally tuned SEF neurons. Our findings are not in agreement with any of the previously suggested models (*Figure 1A–C*). Instead, they support a new sequential decision model (*Figure 1D*). According to this model, at the beginning of the decision two selection processes start independently on the goods and action level. Our data indicate that the SEF activity is part of the action selection process.

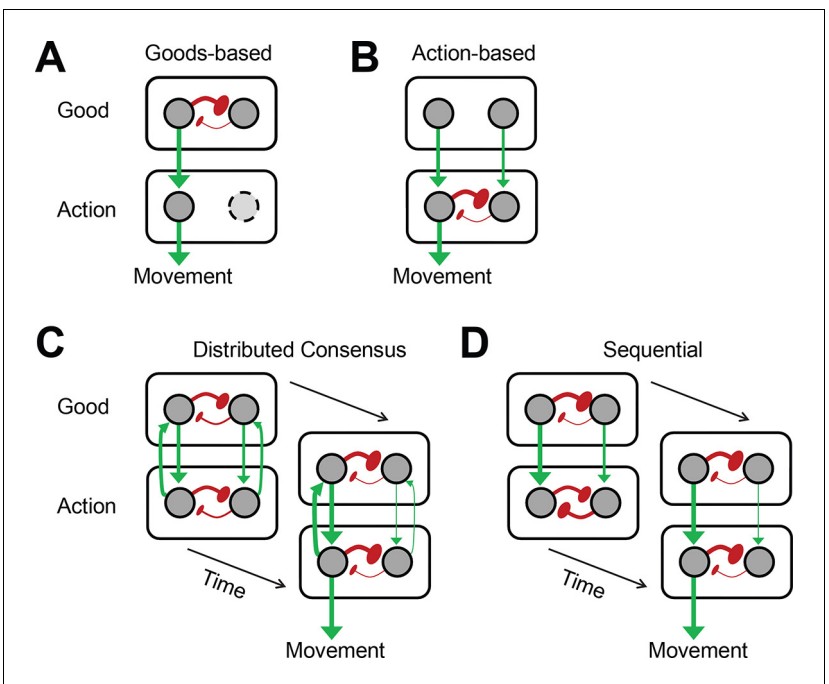

**Figure 1.** Architecture of different decision models. (**A, B**) Goods- and action-based models envision the important selection step during value-based decisions to be either at the value (**A**) or action (**B**) representation stage. (**C, D**) The other two models presume that important selection processes occur at both the value and the action representation stage. However, they differ in their underlying architecture and in the resulting pattern of activity across the network as it unfolds in time. (**C**) The distributed consensus model assumes reciprocal interactions between the value and the action representation. These reciprocal interactions allow the action selection to influence the simultaneous ongoing value selection. The selection of the chosen good and action proceeds therefore in parallel. (**D**) In contrast, the sequential model assumes that there are no meaningful functional reciprocal connections from the action to the value representation. Because of this the action value representations cannot influence the value selection process, which has to finish first, before the action selection can begin. Thus, this decision architecture by necessity implies a sequential decision process. Red arrows indicate excitatory connections. Green buttons indicate inhibitory connections. Thickness of the connection indicates relative strength of the neural activity.

The action selection process receives input from the goods selection. However, due to the absence of recurrent feedback, the goods selection process does not receive input from the action selection process. Once the competition on the goods level is resolved, the value signals for the chosen gamble option increase in strength and the ones for the non-chosen one decrease in strength. This activity difference cascades downward to the action level and determines the outcome of the action selection.

## Results

### Behavior

Two monkeys (A and I) were trained to perform a gambling task in which they chose between two different gamble options with different maximum reward and/or reward probability (*Figure 2A,B*). The maximum and minimum reward amounts were indicated by the color of the target. The portion of a color within the target corresponded to the probability of receiving the reward amount (see experimental procedures). We estimated the subjective value for each target based on the choice preference of the monkeys for all combinations of options (*Figure 2C*) (*Maloney and Yang, 2003*; *Kingdom and Prins, 2010*). The subjective value estimate (referred to in the rest of the paper as 'value') is measured on a relative scale, with 0 and 1 being the least and most preferred option in our set. Consistent with earlier findings (*So and Stuphorn, 2010*), the mean saccade reaction times

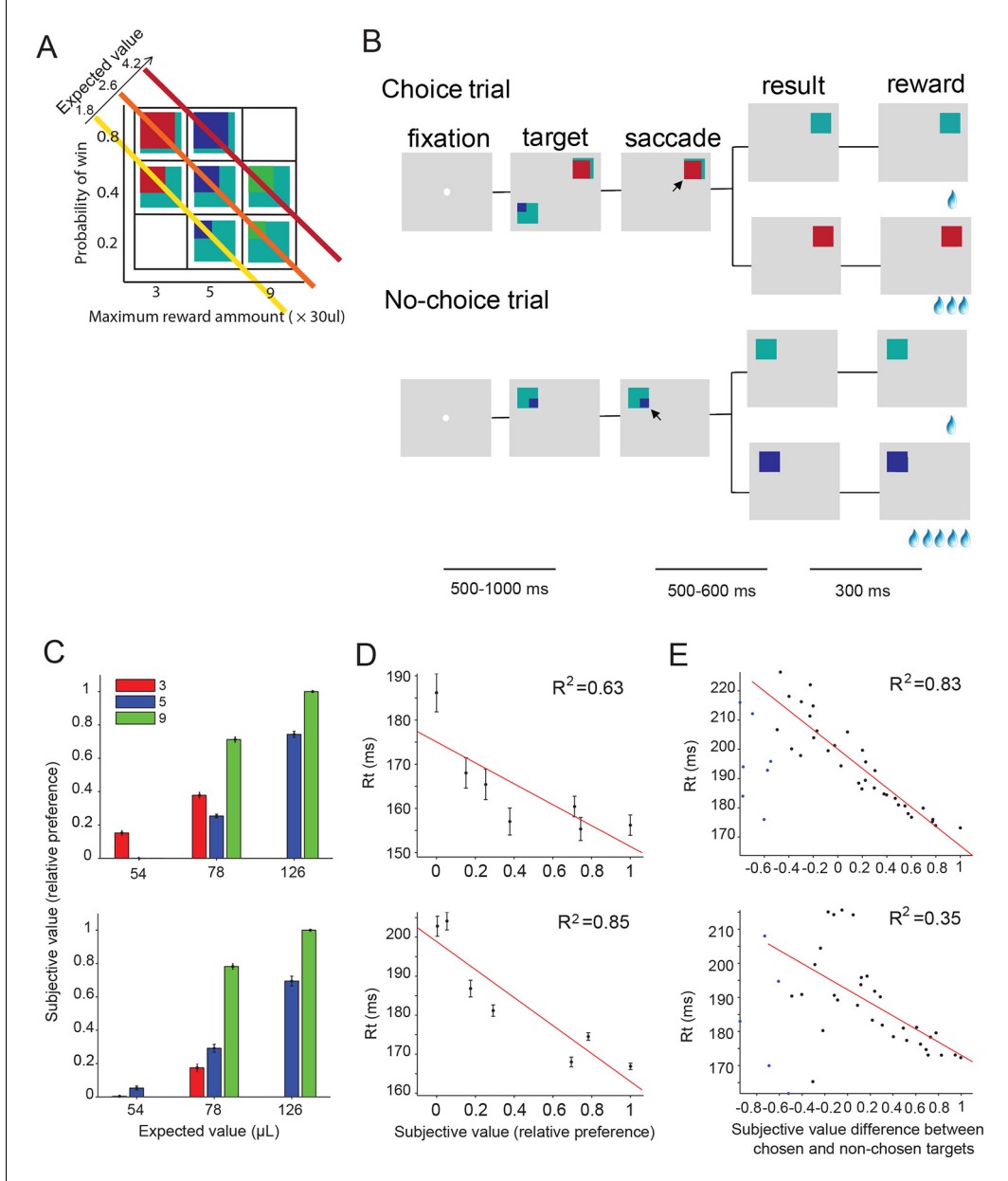

**Figure 2.** Oculomotor gambling task and behavioral results. (**A**) Visual cues used in the gambling task. Four different colors (cyan, red, blue, and green) indicated four different reward amounts (increasing from 1, 3, 5 to 9 units of water, where 1 unit equaled 30 µl). The expected value of the gamble targets along the diagonal axis was the same. For example, the expected value of the bottom right green/cyan target is: 9 units (maximum reward) x 0.2 (maximum reward probability) + 1 unit (minimum reward) x 0.8 (minimum reward probability) = 2.6 units. (**B**) Sequence of events during choice trials (top) and no-choice trials (bottom). The lines below indicate the duration of various time periods in the gambling task. The black arrow is not part of the visual display; it indicates the monkeys' choices. (**C–E**) Behavioral results for monkey A (top) and monkey I (bottom). (**C**) The mean subjective value of the seven gamble options is plotted as a function of expected value. Different colors indicate different amounts of maximum reward. (**D**) The mean reaction times in no-choice trial as a function of subjective value. (**E**) The mean reaction times in choice trial as a function of subjective value differences between chosen and non-chosen targets.

The following figure supplement is available for figure 2:

**Figure supplement 1.** Recording locations in SEF.

during no-choice trials were significantly negatively correlated with the value of the target (*Figure 2D*, monkey B: $t(5) = 8.40$, $p = 0.03$; monkey I: $t(5) = 27.35$, $p = 0.003$). On choice trials, the mean reaction times were significantly correlated with the signed value difference between chosen and non-chosen targets (*Figure 2E*, monkey A: $t(40) = 159.23$, $p < 10^{-14}$; monkey I: $t(38) = 16.18$, $p < 10^{-4}$). Please note that there were a small number of trials with very negative value differences indicating that during these trials the monkey chose a normally non-preferred option. The unusually short reaction time in these trials suggests that the choices were not driven by the normal value-based decision process. These other mechanisms may include history effects, spatial selection bias, express saccades, lapse of attention to the task, and other factors. For this reason, these trials were excluded from the analysis and are marked by a separate color.

## SEF neurons predict chosen option and chosen action sequentially

We recorded 516 neurons in SEF (329 from monkey A, 187 from monkey I, *Figure 2—figure supplement 1*). In the following analysis, we concentrated on a subset of 128 SEF neurons, whose activity was tuned for saccade direction (see experimental procedures).

First, we asked whether SEF activity predicted the chosen gamble option or saccade direction. We performed a trial by trial analysis using linear classification to decode the chosen direction and chosen value from the spike density function with 1 ms temporal resolution. *Figure 3A* shows the classification accuracy across all 128 directionally-tuned SEF neurons. Single neuron activity clearly predicted both chosen gamble option and chosen direction better than chance, but sequentially, not simultaneously. The activity began to predict chosen gamble option around 160 ms before saccade onset and reached a peak around 120 ms before saccade onset, after which it gradually decreased. The activity started to predict saccade direction only around 60 ms before saccade onset. The same pattern is shown by the number of neurons showing significant classification accuracy as a function of time (*Figure 3B*).

This result indicates a sequence of decisions, whereby first an economic good, here a gamble option, is chosen and only later the action that will bring it about. To confirm this finding, we employed an independent information theoretic analysis to study how SEF activity encoded the chosen and non-chosen gamble option, as well as the chosen and non-chosen direction throughout the decision process (*Figure 3C*). We used 106 neurons which were tested with at least 8 out of 12 possible target position combinations. We assumed that the onset of significantly more information about the chosen than the non-chosen variable in the neural firing rate indicated the moment at which the selection process had finished and the choice could be predicted. This moment was reached 66 ms earlier for gamble option information (113 ms before saccade onset; permutation test adjusted for multiple comparison, $p \leq 0.05$) than for saccade direction information (47 ms before saccade onset; permutation test adjusted for multiple comparisons, $p \leq 0.05$). Thus, the onset and timing of the information representation in SEF is consistent with the results of the classification analysis and also indicates a sequential decision process.

In our gamble task, the monkey was free to indicate his choice as soon as he was ready. Because of this design feature, the saccade onset is likely to be closer aligned with the conclusion of the decision process than target onset. The fact that reaction time reflected chosen value and value difference (i.e. choice difficulty), as indicated in *Figure 2*, also supports this idea. We analyzed therefore the neural activity aligned on movement onset, because it likely reflects the dynamic of the decision process more accurately. The analysis of the neural activity aligned on target onset further confirms this conclusion (*Figure 3—figure supplement 1*).

## SEF neurons reflect the value of both choice options in an opposing way

Our findings indicated that SEF neurons show signs of a sequential decision process, whereby first a desired economic good is chosen and only then the action that is necessary to obtain the good. Next, we investigated the neural activity in the SEF neurons more closely to test if the SEF only reflects the outcome of the decision, or whether it also reflects one or both of the selection steps. Specifically, we searched for opposing contributions of the two choice options to the activity of SEF neurons, which would indicate a competitive network that could select a winning option from a set of possibilities.

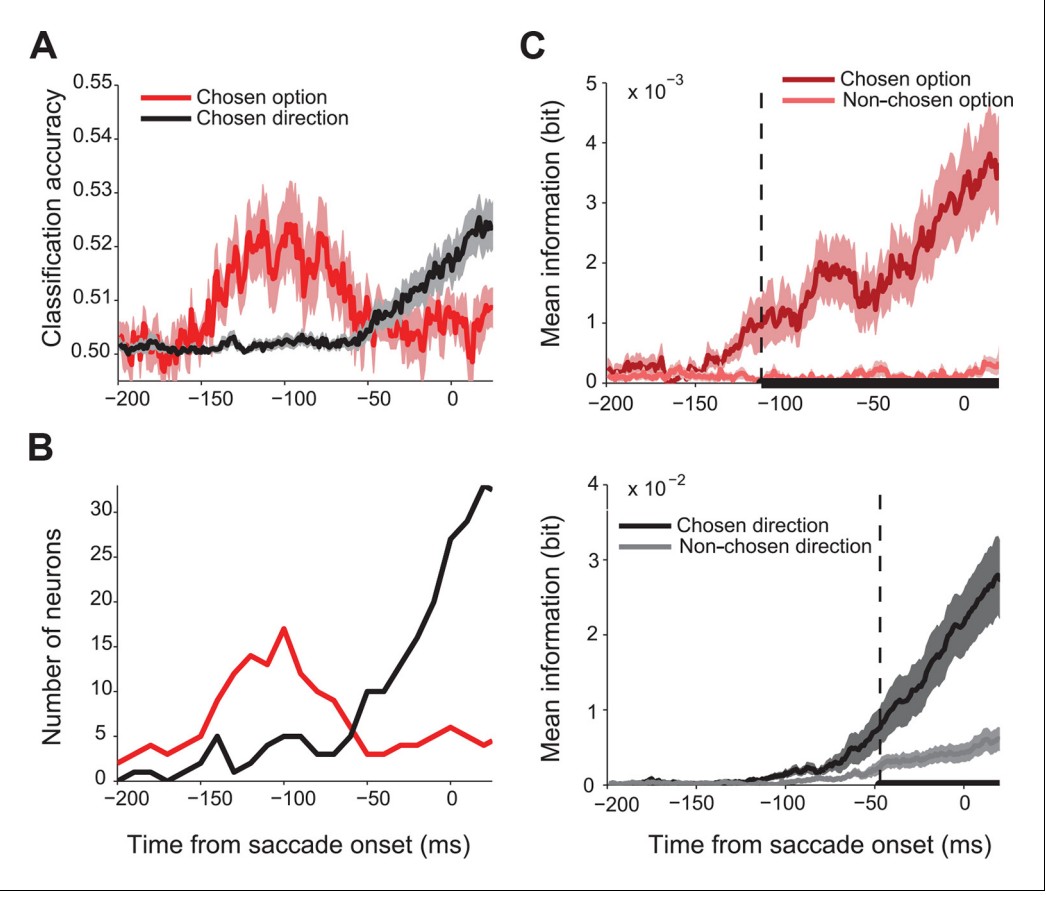

**Figure 3.** Time course of chosen gamble option and saccade direction representation in SEF. (**A**) Significant classification accuracy for chosen gamble option (red) and chosen saccade direction (black) across 128 neurons. We excluded values that were not significantly different from chance (permutation test; p≤0.05). (**B**) Number of neurons showing significant classification accuracy for chosen gamble option (red) and chosen saccade direction (black). (**C**) Average mutual information between SEF activity and chosen and non-chosen gamble option (top panel; dark and light red) and saccade direction (bottom panel; dark and light grey). The time period when the amount of information about chosen and non-chosen option/direction was significantly different (paired t-test adjusted for multiple comparisons, p≤0.05) are indicated by the thick black line at the bottom of the plots. The onset of a significant difference is indicated by the vertical dashed line. SEF, supplementary eye field.

The following figure supplement is available for figure 3:

**Figure supplement 1.** Time course of value and saccade direction representation in SEF aligned on target onset.

The directionally tuned SEF neurons represented the value of targets in the preferred direction (PD) (**Table 1**). The PD is the saccade direction for which a neuron is maximally active, irrespective of reward value obtained by the saccade. We estimated each neuron's PD using a non-linear regression analysis of activity for saccades to all four possible target locations. We defined here PD as the target direction that is closest to the estimated PD. **Figure 4A,B** shows the activity of the SEF neurons during no-choice trials, that is, when only one target appears on the screen. Although the SEF neurons are strongly active for PD targets and show value-related modulations (**Figure 4A**), they are not active for saccades into the non-preferred direction (NPD), independent of their value (**Figure 4B**; regression coefficient = 0.013, t(5) = 1.324, p=0.243). The SEF neurons encode, therefore, the value of saccades to the PD target, confirming previous results (**So and Stuphorn, 2010**).

There are a number of subtypes of value signals that are associated with actions, such as saccades. These signals are related to different stages of the decision process (**Schultz, 2015**; **Stuphorn, 2015**). First, there are signals that represent the value of the alternative actions irrespective of

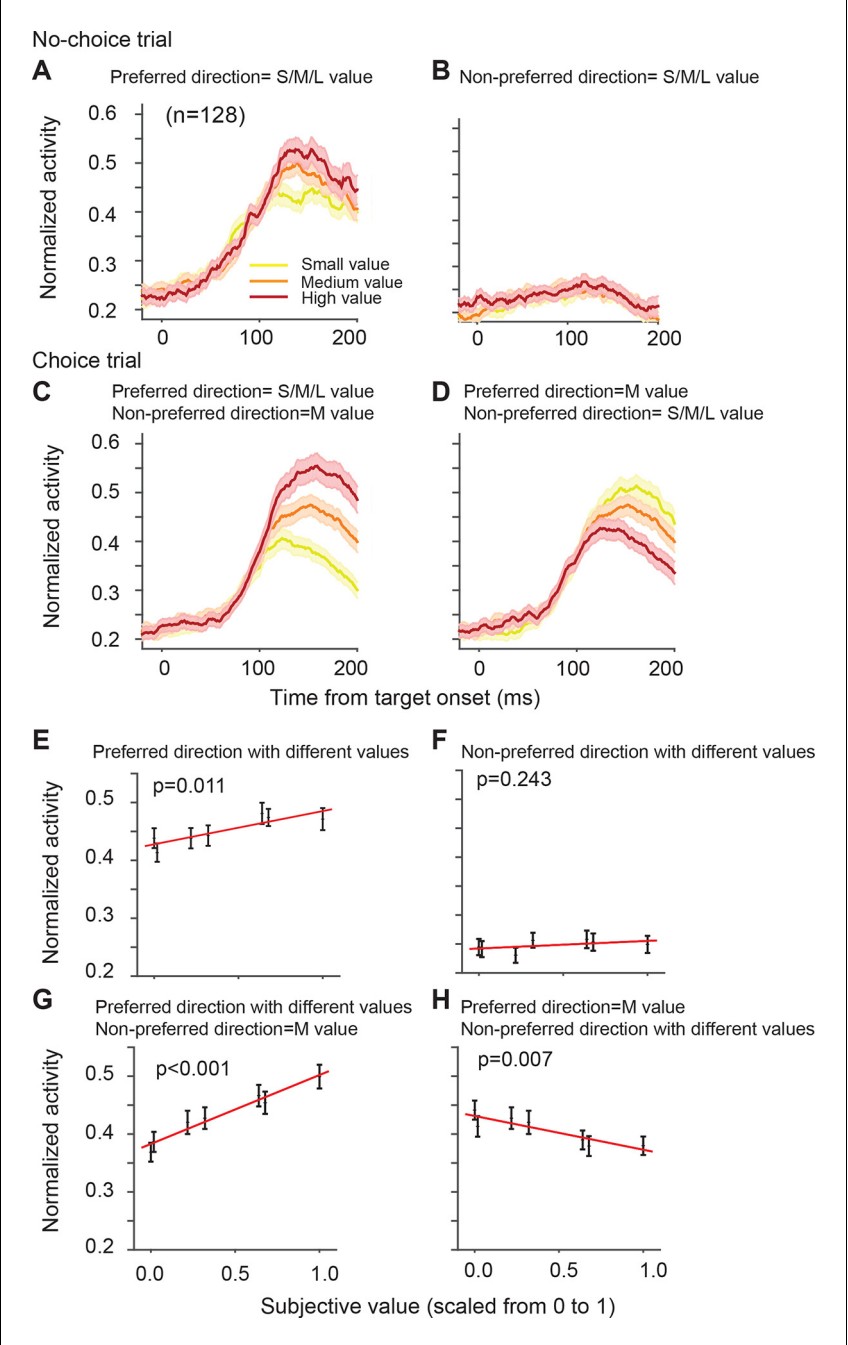

**Figure 4.** SEF neurons represent the difference in action value associated with targets in the preferred and non-preferred direction. The neural activity of 128 directionally tuned SEF neurons was normalized and compared across trials with different values of targets in the preferred or non-preferred direction. (**A**) The neural activity in no-choice trials, when the target was in the preferred direction. (**B**) The neural activity in no-choice trials, when the target was in the non-preferred direction. (**C, D**) The neural activity in choice trials. To visualize the contrasting effect of targets in the preferred or non-preferred direction on neural activity, the value of one of the targets was held constant, while the value of the other target was varied. Activity was sorted by target value, but not by saccade choice. (**C**) The neural activity, when the value of the target in the preferred direction varied, while the value of the target in the non-preferred direction was held constant at a medium value. (**D**) The neural activity, when the value of the target in the non-preferred direction varied, while the value of the target in the preferred direction was held constant at a medium value. The color of the spike density histograms indicates the target value [high value = 6–7 units (red line); medium value = 3–5 units (orange line); low value = 1–2 units (yellow line)]. (**E-H**) The regression analysis corresponding to (**A-D**). A t-test was used to determine whether the regression

*Figure 4 continued on next page*

*Figure 4 continued*

coefficients were significantly different from 0. The regression coefficients, confidence intervals, t-values, and p-values are listed in *Table 1*. SEF, supplementary eye field.

The following figure supplements are available for figure 4:

**Figure supplement 1.** SEF neurons represent the difference in value of targets in the preferred and non-preferred direction.

**Figure supplement 2.** SEF neurons represent the difference in value of targets in the preferred (PD) and non-preferred direction (NPD) independent of the chosen saccade direction.

**Figure supplement 3.** SEF neurons represent the difference in action value associated with targets in the preferred and non-preferred direction.

**Figure supplement 4.** Neural activity modulated by the relative angle and position of the two targets.

the choice. These signals represent the decision variables on which the decision process is based and are commonly referred to as 'action value' signals. Second, there are signals that encode the central step in the decision process, namely the comparison between the values of the alternative actions. Such 'relative action value' signals represent a combination of different 'action value' signals. They should be positively correlated with the action value of one alternative and negatively correlated with the action value of the other alternatives. Third, there are signals that indicate the value of the chosen action. These 'chosen action value' signals represent the output of the decision process. Single target trials do not allow to distinguish between these functionally very different signals, but choice trials do.

On their own, NPD targets did not evoke neural activity. However, the value of NPD targets clearly modulated the response of the SEF neurons to the PD targets in choice trials (*Figure 4D*; *Table 1*). To isolate the effect that targets in the two directions have on the neural activity, we first held the value of the NPD target constant at a medium amount and compared the SEF population activity across trials with PD targets of varying value (*Figure 4B*). The neural activity clearly increased with the value of the PD target (regression coefficient = 0.119, $t(5) = 10.629$, $p<0.001$; *Figure 4E*). Next, we held the value of the PD target constant at a medium value and compared the population activity across trials with NPD targets of varying value (*Figure 4C*). The neural activity clearly decreased with the value of the NPD target (regression coefficient=-0.058, $t(5)=-4.345$, $p=0.007$; *Figure 4F*). Thus, the SEF neurons represented a relative action value signal. The value of the PD target influences the SEF activity about twice as large as the value of the NPD target. This means that

**Table 1.** Average value effect on neural activity across all directional SEF neurons. The upper two rows show the effect of preferred and non-preferred direction target value on normalized neuronal activity in no-choice trials, and the lower two rows show their effect in choice trials. Within each set, the upper row ($V_{PD}$) shows the effect of the preferred direction target value on normalized neural activity, whereas the lower row shows the effect of the non-preferred direction target value on normalized neural activity. Significance was calculated using a t-test, which shows whether the regression coefficient is significant difference from zero. The analysis corresponds to the results presented in *Figure 4*.

| All neurons (n = 128) | | Regression coefficient | Lower confidence bound | Upper confidence bound | t(5) | p |
|---|---|---|---|---|---|---|
| No-choice | $V_{PD}$ | 0.057 | 0.020 | 0.095 | 3.945 | 0.011 |
| | $V_{NPD}$ | 0.013 | -0.012 | 0.039 | 1.324 | 0.243 |
| Choice | $V_{PD}$ | 0.119 | 0.090 | 0.148 | 10.629 | <0.001 |
| | $V_{NPD}$ | -0.058 | -0.093 | -0.0238 | -4.345 | 0.007 |

SEF, supplementary eye field.

the SEF neurons do not encode the exact value difference. Nevertheless, the opposing influence of the targets indicates that SEF represents the essential step in decision-making, namely a comparison of the relative value of the available actions. All these effects were present well before saccade onset (*Figure 4—figure supplement 1*) and did not depend on the chosen saccade direction (*Figure 4—figure supplement 2*). Similar activity pattern can also be observed if pooling together all task related neurons (N=353, *Figure 4—figure supplement 3*). In contrast, the neurons were not significantly influenced by the relative spatial location of each target (*Figure 4—figure supplement 4*).

We used a regression analysis to further quantify the relative contribution of the chosen and non-chosen gamble option and saccade to the neural activity during the decision. In modeling the neural activity in choice trials, we used each neurons' activity in no-choice trials as a predictor of its response in choice trials. Specifically, we modeled the neural activity as a weighted sum of the activity in no-choice trials for saccades to targets with the same gamble option or direction as the chosen and non-chosen targets in the choice trials. The strength of the coefficients is a measure of the relative influence that each target has on the neural activity in a particular time period during the decision process. *Figure 5* shows the time course of the coefficient strength for the chosen and non-chosen target when we sorted trials either by gamble option or saccade direction. In both cases, the correlation coefficients for the two targets were initially of equal value, indicating that the SEF

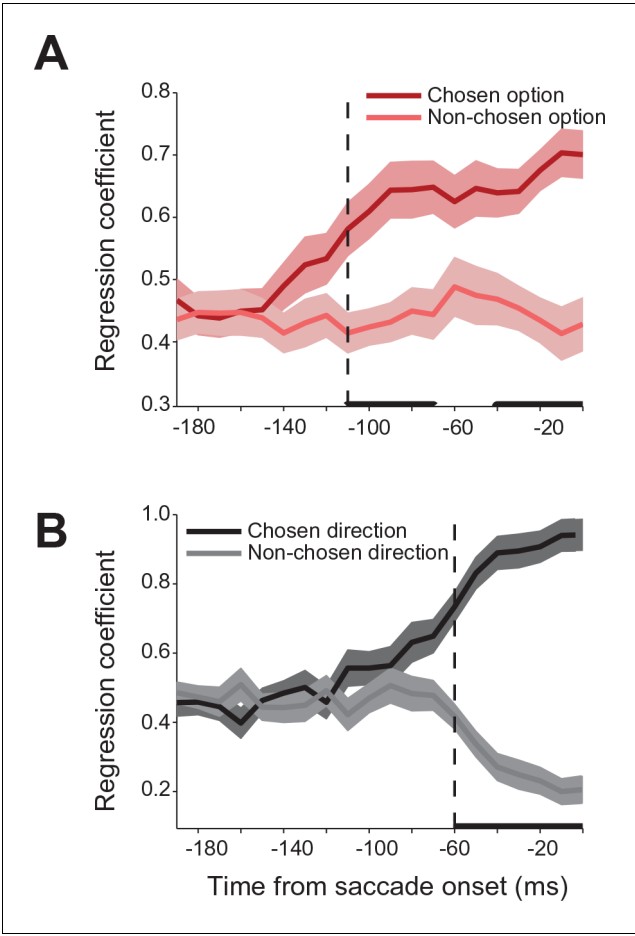

**Figure 5.** Relative influence of chosen and non-chosen target on SEF activity. (**A**) Regression coefficients for chosen and non-chosen gamble options (dark and light red). (**B**) Regression coefficients for chosen and non-chosen saccade directions (dark and light grey). Time periods in which the regression coefficients for chosen and non-chosen option/direction are significantly different (paired t-test adjusted for multiple comparisons, p≤0.05) are indicated by a thick black line. The onset of a significant difference is indicated by a vertical dashed line. All panels are aligned on saccade onset. The shaded areas represent SEM. SEF, supplementary eye field; SEM, standard error of the mean.

neurons reflected each target equally during this time period. However, 110 ms before saccade onset (permutation test adjusted for multiple comparisons, p≤0.05) the strength of the chosen gamble option coefficient started to rise, while the non-chosen gamble coefficient strength stayed the same. Later in the trial, 60 ms before saccade onset (permutation test adjusted for multiple comparisons, p≤0.05) the coefficient strength for the chosen saccade direction increased. Simultaneously, the coefficient strength for the non-chosen saccade direction decreased. The results of the regression analysis allow a number of conclusions about the mechanism underlying decision-making and the role of SEF in it. First, the results confirm the findings of the decoding and encoding analysis (*Figure 3*) and indicate that the decision process involves a sequence of two different selection processes. Second, the opposing pattern of influence on neural activity in the case of saccade direction suggests that the latter action selection step could involve at least partially the SEF. The mechanism underlying this selection involves competition between the action value signals associated with the two saccade targets. Thus, the increasing influence of one target leads simultaneously to a decreasing influence of the other target. However, the neural activity associated with the earlier gamble option selection step does not show such a pattern of competing influences. Instead, the influence of the chosen option just increases. That indicates that the choice of an economic good does not involve competitive interactions between SEF neurons. Instead, only the output of the gamble option selection is represented in SEF. This signal could reflect input from other brain regions.

## Relative action value map in SEF reflects competition between available saccade choices

Each directionally tuned SEF neuron represents the relative action value of saccades directed toward its PD. Together these neurons form a map encoding the relative value of all possible saccades during the decision process. Our analysis of the activity pattern in individual neurons suggested that the action selection relied on competition between different relative action value signals. In that case, the relative action value map in SEF should contain different groups of neurons that represent the competing relative action values of the two saccade choices. If the activities of these two groups of neurons indeed reflect inhibitory competition, the selection of a particular action should lead to increased activity of the neurons representing the chosen and decreased activity in the neurons representing the non-chosen saccade. Furthermore, the inhibition that the winning neurons can exert on the losing neurons should depend on their relative strength. We should therefore see differences in the dynamic of the neural activity within the relative action value map if we compare trials with small or large value differences.

To reconstruct the SEF relative action value map, we combined the activity of all directionally tuned neurons in both monkeys (*Figure 6*). We sorted each SEF neuron according to its PD and normalized their activity across all trial types (choice, no-choice trials). We then smoothed the matrix by linear interpolation at a bin size of 7.2° and plotted the activity. For each successive moment in time, the resulting vector represented the relative action value of all possible saccade directions, because all task-relevant saccades were equidistant to the fixation point. The succession of states of the map across time represented the development of relative action value-related activity in SEF over the course of decision making.

In choice trials, activity started to rise in two sets of neurons (*Figure 6B*). One was centered on the chosen target (indicated by the red dot), while the other one was centered on the non-chosen target (indicated by the black dot). The initial rise in activity was not significantly different between choice and no-choice trials (onset time on no-choice trial: 44 ms, choice trial: 40 ms; permutation test adjusted for multiple comparisons, p≤0.05). However, there was a longer delay between the initial rise in activity and saccade onset (onset time on no-choice trial: 141 ms before saccade onset, choice trial: 185 ms before saccade onset; permutation test adjusted for multiple comparisons, p≤0.05), in keeping with the fact that reaction times were longer when the monkey had to choose between two response options (*Figure 2E*). At the beginning, the activity associated with both possible targets was of similar strength, but 70 ms before saccade onset (permutation test adjusted for multiple comparisons, p≤0.05), a significant activity difference developed between the two sets of cells that predicted which saccade would be chosen. The activity centered on the chosen saccade became much stronger than the one centered on the non-chosen saccade. This differentiation reflected the decision outcome within the SEF relative action value map.

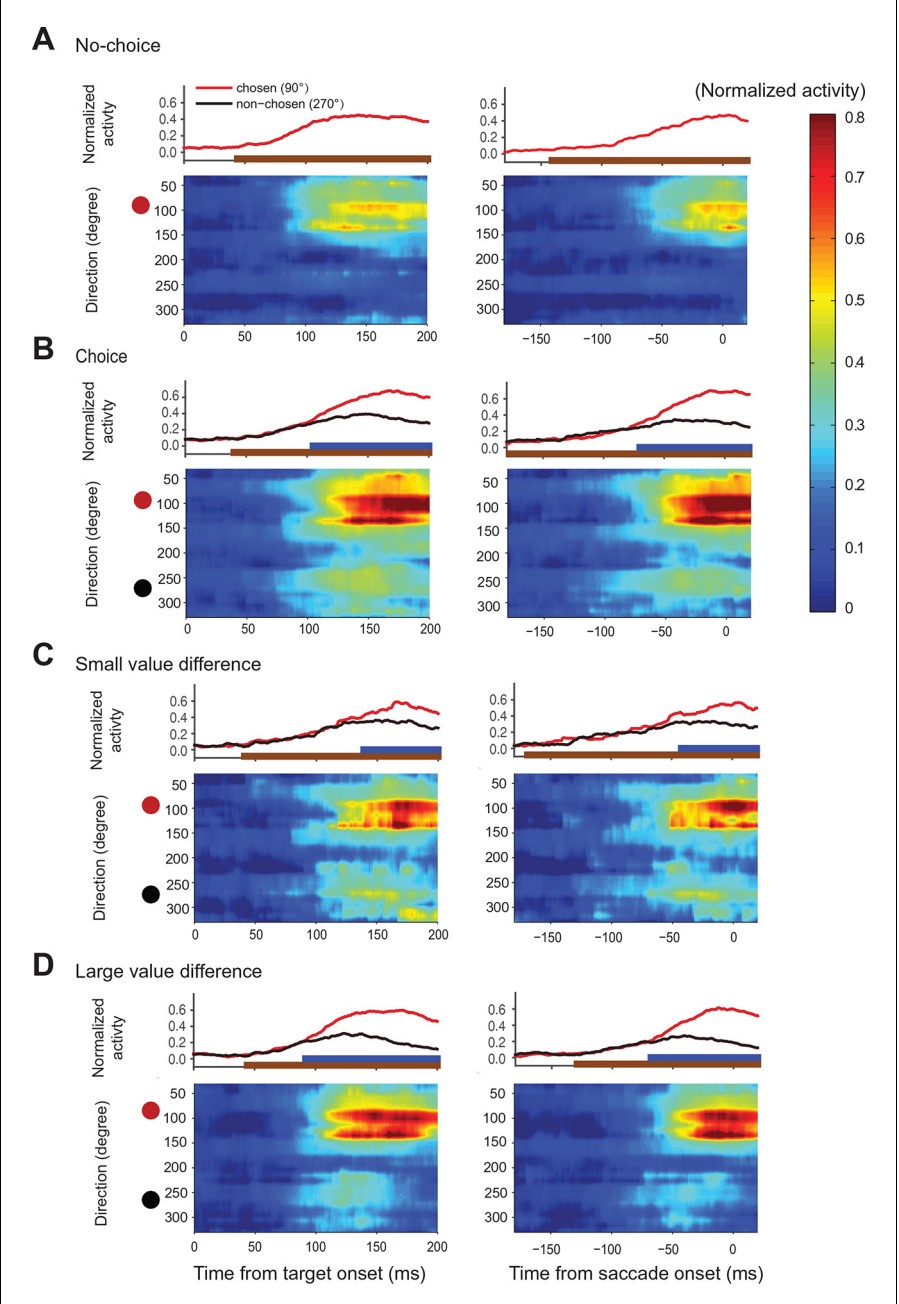

**Figure 6.** Action value maps showing population activity in SEF during decision making. Each neuron's activity was normalized across all trial conditions. The maps in the left column are aligned on target onset and the panels in the right column on saccade onset. In each map, horizontal rows represent the average activity of cells whose preferred direction lies at a given angle relative to the chosen target (red circle on left). Color indicates change in normalized firing rate relative to the background firing rate (scale on the right). (**A**) Population activity during no-choice (**A**) and choice trials (**B**). Population activity in choice trials divided into trials with small (**C**) and (**D**) large value differences between the reward options. The subplots above the action value maps show the time course of the neural activity associated with the chosen (45–135°) and non-chosen (225–315°) target. The brown lines underneath show the time when population activities were significantly different than the baseline (permutation test adjusted for multiple comparisons). The blue lines underneath show the time when the neural activities associated with the chosen target were significantly different from those associated with the non-chosen target (permutation test adjusted for multiple comparisons). SEF, supplementary eye field.

*Figure 6 continued on next page*

*Figure 6 continued*

The following figure supplement is available for figure 6:

**Figure supplement 1.** Action value maps for trials with variable choices.

The chosen saccade was often also the one with the larger value. We therefore performed a separate analysis of those choice trials were the monkeys made different choices for the same pair of gambles, which allowed us to differentiate between the representation of action value and choice. A comparison of the trials when the larger ('correct') or smaller ('error') value targets were chosen shows a strong increase of neural activity for the chosen target regardless of its average value (*Figure 6—figure supplement 1*). This confirmed that neural activity in SEF represented not only the relative action value of the competing saccades, but also the final choice.

We hypothesized that the action selection process is driven by competition between the action values of the two targets. If that were true, we would expect the reduction in non-chosen saccade related activity to be less pronounced and to occur later because of weaker inhibition, when the differences in action values were less pronounced. We divided therefore the choice trials into two groups with small (value difference smaller than 0.4) and large value differences (value difference larger or equal to 0.4), while controlling that the mean chosen value in both conditions was the same (*Figure 6C,D*). As predicted, the neural activity associated with the non-chosen target was stronger and lasted longer, when the value differences were small (onset of significant activity difference for chosen target and non-chosen target was 68 ms before saccade onset for small value difference trial and 42 ms before saccade onset for large value difference trials; permutation test adjusted for multiple comparisons, $p \leq 0.05$, *Table 2*). This longer lasting activity was consistent with the longer reaction time for smaller value difference trials (*Figure 2E*). In contrast, the activity for the chosen target was weaker, when the value differences were small, especially early on (100–150 ms after target onset). The stronger activity associated with the non-chosen target in these trials was likely better able to withstand the competition of the activity associated with the chosen target and in turn reduced this activity more strongly.

On choice trials, there is a simultaneous onset of activity in the two areas of the relative action value map that correspond with the location of the two target options (*Figure 6B*). Throughout the task, there is a robust representation of both options maintained in SEF, even after the divergence of activity that indicates the chosen option and action (*Figure 4C,D*; *Figure 6B,C*). However, during the initial rise in activity the SEF population does not indicate the value of the target in its PD. At the time, when the neurons start to differentiate their activity according to the value of the target in the preferred direction they also reflect the value of the target in the NPD. This can be seen very clearly in *Figure 4D* showing the activity of SEF neurons for PD targets of medium value. Depending on the value of the NPD target, the activity starts to change ~110–120 ms after target onset. However, it took this much time for the SEF neurons to indicate value even during no-choice trials, when there was no competing target. It seems therefore that the SEF neurons always indicate relative action value. There is no time period in which two populations of SEF neurons represent the absolute action value of a target independent of the value of any competing target. Nevertheless, there is clear evidence of a succession of an initial undifferentiated state to a more and more differentiated state in which the influence of the chosen action value on the neuronal activity increases and the influence of the non-chosen one decreases (*Figure 4D*; *Figure 6B,C*). This indicates a dynamic process as would be expected by a decision mechanism driven by competition via inhibitory interactions.

Our results therefore support the idea that an ongoing process of inhibitory competition underlies the action selection. SEF neurons might directly participate in this action selection process, or at least reflect it.

## Instantaneous changes in SEF activity state space reflect decision process

So far, all analysis have been performed using individual SEF neurons or comparisons of specific subsets of neurons. However, the decision process should also manifest itself in the dynamic changes in the instantaneous activity distribution across the entire SEF population. To study how the SEF

**Table 2.** The onset times in time-direction maps. The first main column shows the onset time calculated from trials aligned on target onset and the second main column shows the onset time calculated from trials aligned on saccade onset. Within each main column, the first minor column shows the time when the neural activity was significantly different from background activity (-20 to 0 before target onset). The second minor column shows the time when the neural activity represented the choice. In no-choice trials, this corresponds to the time when the activity of neurons with a preferred direction within ± 30° of the target was significantly different from the activity of neurons where no target was presented (the neurons with preferred direction within 240–300°). For choice trial, it corresponds to the time when the activity for the chosen target was significant different form the activity for the non-chosen target (in both cases the neurons with preferred direction within ± 30° of their respective target). A permutation test with multiple comparison correction was used to calculate the onset times.

| | Time from target onset | | Time from saccade onset | |
|---|---|---|---|---|
| | Activity vs background | Chosen vs non-chosen | Activity vs background | Chosen vs non-chosen |
| No-choice | 44 ms | | -141 ms | |
| Choice | 40 ms | 105 ms | -185 ms | -70 ms |
| Choice (dV>=0.4) | 44 ms | 92 ms | -129 ms | -68 ms |
| Choice (dV<0.4) | 41 ms | 139 ms | -169 ms | -42 ms |

population dynamically encodes the task variables underlying the monkeys' behavior, we analyzed the average population responses as trajectories in the neural state space (*Yu et al., 2009*; *Shenoy et al., 2011*). As the previous analyses show, the activity pattern of individual SEF neurons during the decision process is not completely independent from each other, but follows particular pattern. The movement of SEF activity state trajectories in a lower-dimensional sub-space captured therefore most of the relationship between the SEF activity state and the behaviorally relevant task variables (*Figure 7*). We estimated this task-related subspace by using linear regression to define three orthogonal axis: chosen saccade direction along the horizontal and vertical dimension, and value of the chosen option (*Mante et al., 2013*).

First, we compare only trajectories for saccades in two different directions and three different value levels in a simplified two-dimensional space spanned by the value and the vertical movement direction axis (*Figure 7A*). The trajectories of upward and downward saccades (as indicated by the different color) are clearly well separated along the direction axis. In addition, the trajectories for different chosen value are also separate from each other along the value axis (as indicated by different line style). Thus, the trajectories move in an orderly fashion with respect to the two task-related axis. As a result of the separation across both axis, the trajectories reach six different points in state space, when the respective saccade is initiated.

Similarly, in the full three-dimensional (3-D) task space, the trajectories for all four directions and the different chosen values are also well separated from each other and do not converge (*Figure 7B*). As a result, the trajectories reach different positions in 3-D task space at the moment of saccade initiation, and their distance is significantly correlated with the difference in chosen value for all four different saccade directions (*Figure 7C*). Our previous analysis suggests that the neurons encoding the relative action value of saccades in different direction compete through mutual inhibition. This mutual inhibition should change the direction and endpoint of the different trajectories, so that they should not only depend on the saccade direction and subjective value of the chosen target, but also on the value of the non-chosen target. To test this, for a fixed direction and chosen value, we computed the distance between the trajectory with the largest non-chosen value and all other trajectories with decreasing non-chosen values. The regression analysis shows that, when saccade direction and chosen value is fixed, the distance between trajectories is significantly modulated by non-chosen value (*Figure 7D*). The larger the non-chosen value difference is the further apart the trajectories are when the saccade is initiated. This can also be observed by comparing population activity trajectories in the state space (*Figure 7—figure supplement 1*). Thus, the trajectories in

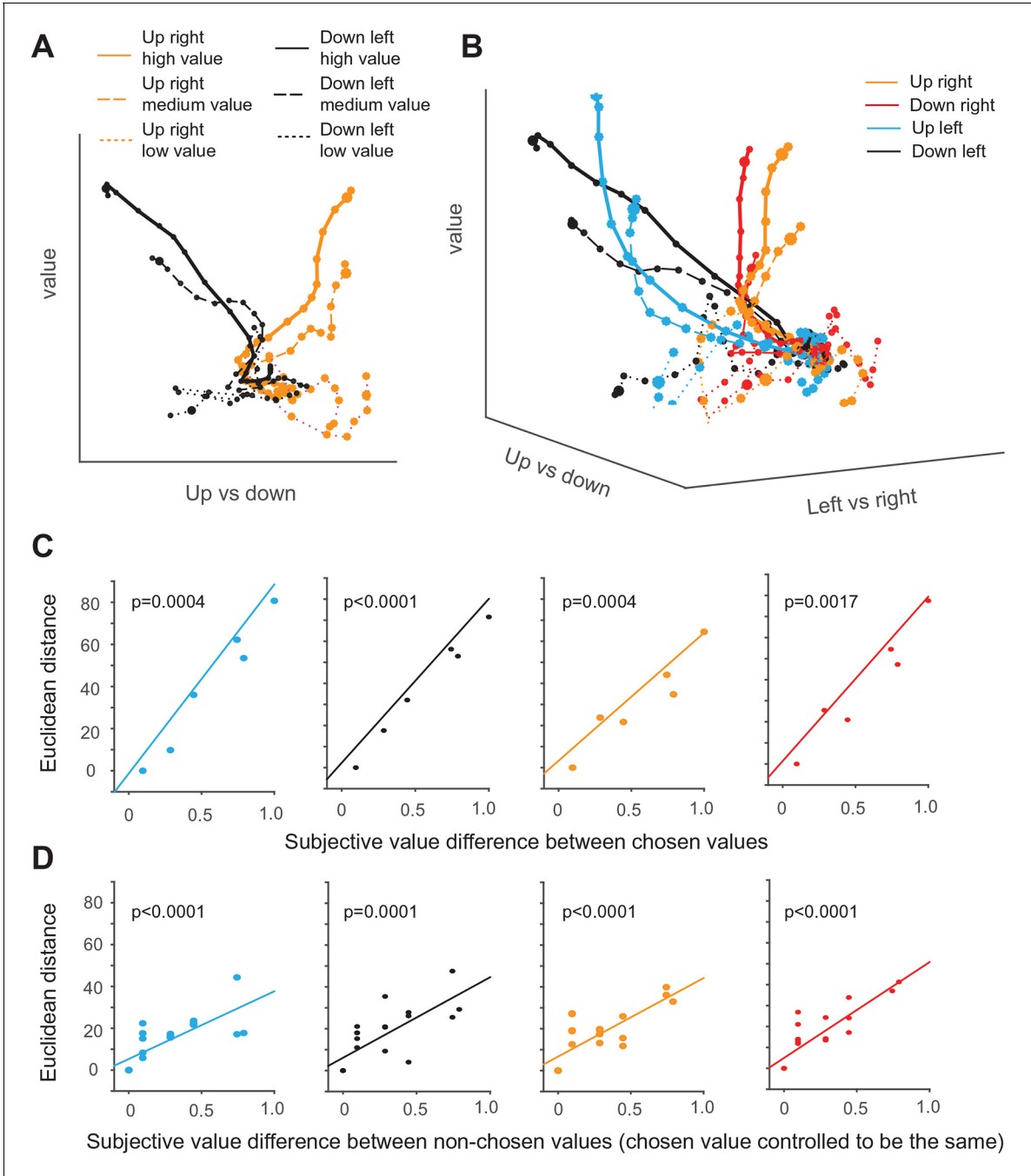

**Figure 7.** Dynamics of SEF population activity trajectories in state space during decision making. The average population response for a given condition and time period (10 ms) is represented as a point in state space. Responses are shown from 200 ms before to 10 after saccade onset. The time of saccade initiation is indicated by the larger dot. The four different chosen saccade directions are indicated by different colors (up right: red; down right: orange: up left: black; down left: blue) and the value of the chosen target by line style (high value (value>=0.7): solid line, medium value (value<0.7 and value >0.3): dashed line, low value dotted line). (A) Trajectories of up-left and down-right movement in value and horizontal (left/right) subspace for three different values. (B) Trajectories of movements in value and action subspace. (C) The effect of the chosen option value on the state space trajectory at saccade onset. The subjective value of each chosen option was measured relative to the option with the smallest chosen value. The Euclidian distance in 3-D task space between the state vectors of each pair of chosen options increased as a function of their difference in subjective value. The significance of the relationship between difference in Euclidian distance and value was tested using a regression analysis (t-test; the p-value indicates the probability that the regression slope is significantly different from zero). (D) The effect of the non-chosen option value on the state space trajectory at saccade onset. For trajectories with fixed saccade direction and chosen option value, the difference in Euclidian distance increased as a function of difference in subjective value of each non-chosen option relative to the option with the largest non-chosen value. SEF, supplementary eye field.

*Figure 7 continued on next page*

*Figure 7 continued*

The following figure supplements are available for figure 7:

**Figure supplement 1.** Dynamics of SEF population activity trajectories in state space for choices of the down-left target.

**Figure supplement 2.** Fraction of variance in state vector position explained by task-related axes of chosen direction (pink and purple) and chosen value (green).

state space reflect both the chosen and the non-chosen target values. In this context, no-choice trials could be considered as choice trials with zero non-chosen value. In this case, one would expect the trajectory to be similar to the trials with large chosen and small unchosen value. However, this is not the case (*Figure 7—figure supplement 1*). Although trajectories for no-choice trials are also modulated by both value and direction, they always reach a point along both the value and direction axis that is less extended than during choice trials.

Lastly, we asked whether the sequential value and action selection can be observed in the state space analysis. An indication of this can be seen in *Figure 7A*. The trajectories first start to separate along the value axis, before they separate along the direction axis. Consistent with this observation and the single neuron analysis, the variance explained by value axis increased earlier than the variance explained by saccade direction axis (*Figure 7—figure supplement 2*).

## Discussion

Our results indicate that SEF represents the relative action value of all possible saccades, forming a relative action value map. During value-based decisions, the SEF population first encodes the chosen gamble option and only later the chosen direction. Our data suggest that neural activity in SEF reflects the action selection, the second step in the decision process. This selection process occurs likely through a process of competitive inhibition between groups of neurons carrying relative action value signals for different saccades. This inhibition could occur locally between SEF neurons, either through mutual inhibition between different SEF neurons within the relative action value map, or through a global pool of inhibitory neurons that receive input from excitatory neurons in SEF (*Schlag et al., 1998*; *Wang, 2008*; *Nassi et al., 2015*). Alternatively, the neural activity in SEF and the competitive process manifested by it could reflect shared signals within the more distributed action selection network that SEF is part of. Similar action value signals have been reported in lateral prefrontal cortex (*Matsumoto et al., 2003*; *Wallis and Miller, 2003*), anterior cingulate cortex (*Matsumoto et al., 2003*), and basal ganglia (*Samejima et al., 2005*; *Lau and Glimcher, 2008*). Of course, it is also possible that the action selection involves both interactions between neurons in a larger network and local inhibitory interactions. Future perturbation experiments will be required to test whether SEF plays a causal role in decision making and also if the relative action value encoding is at least partly the result of local inhibitory mechanism or whether it reflects only input from connected brain regions. Independent of these considerations, our results allow us to draw some conclusions about the basic functional architecture of decision making in the brain. Specifically, they invalidate a number of previously suggested decision models and instead support a new sequential model of decision making.

Currently, three major hypotheses about the mechanism underlying value-based decision making have been suggested: the goods-based model (*Figure 1A*), the action-based model (*Figure 1B*), and the distributed consensus model (*Figure 1C*). Our gamble experiment design allows us to test these models by dissociating the value selection process from the action selection process. Due to the uncertainty of reward for each individual gamble, the large number of the gamble option pairs, and the fact that each gamble option pairing could be presented in multiple spatial configurations, the task design prevent the subjects from making direct associations between the visual representation of the gamble options and action choice. Therefore, the task required on each trial a good-based, as well as an action-based selection.

Our findings support none of the previously suggested models. First, the pure good-based model of decision making would predict that action-related representations are downstream of the decision

stage and should therefore only represent the decision outcome (i.e. the chosen action; see *Figure 1A*). However, we found evidence for competition between relative action value-encoding neurons in SEF that is spatially organized, that is, in an action-based frame of reference (*Figure 4*). This fact, together with the fact that we find activity corresponding to both response options (*Figure 6*) clearly rules out the pure good-based model (*Figure 1A*). Second, the pure action-selection model B would predict that the competition would only happen in the action value space. In *Figure 1B*, this is indicated by the absence of inhibitory connections between the nodes representing the reward options (or goods). This model therefore predicts that information about chosen saccade direction should appear simultaneously with or even slightly earlier than information about chosen reward option, since the selected action value signal contains both direction and value information. However, this prediction is contradicted by our observation that the chosen value information is present earlier than the chosen action information (*Figure 3*, *Figure 5* and *Figure 7*). Thus, there is a moment during the decision-making process (100–50 ms before saccade onset), when the SEF neurons encode which option is chosen, but not yet, what saccade will be chosen. This could clearly never happen in an action-selection model of decision-making. Lastly, the distributed consensus model (*Figure 1C*) suggests strong recurrent connections from the action selection level back to the option selection level. The reciprocal interaction should lead to a synchronization of the selection process in both stages and the chosen gamble option and chosen action should be selected simultaneously. This prediction is clearly not supported by our findings, given the robust 100 ms time difference in the onset of chosen option and direction information.

Instead, our data are most consistent with a model that predicts selection both on the option and the action representation level, but with asymmetric connections between them, so that the option selection level influences the action selection level, but not vice versa. This is the sequential model of decision making (*Figure 1D*). According to this model, value-based decisions require two different selection processes within two different representational spaces. First, a preferred option has to be chosen within an offer or goods space by comparing their value representations (*Padoa-Schioppa, 2011*). Second, within an action space the response has to be chosen that most likely will bring about the preferred option. A similar sequence has been found, when initially only information relevant for the selection of an economic good is provided, and information relevant for the selection of an action is only given after a delay (*Cai and Padoa-Schioppa, 2014*). Here, we show that this sequence is obligatory, since it even occurs, when information guiding the selection of goods and actions is given simultaneously. This suggests that these two selection steps are related, but functionally independent from each other and involve different brain circuits. This explains why evidence for decision-related neural activity has been found at both selection stages (*Shadlen et al., 2008*; *Cisek and Kalaska, 2010*; *Wunderlich et al., 2010*; *Padoa-Schioppa, 2011*; *Cisek, 2012*). A similar separation between stimulus categorization and action selection has been also found in other decision processes (*Schall, 2013*). The competition between subjective value representations takes most likely place in OFC (*Padoa-Schioppa and Assad, 2006*) and vmPFC (*Wunderlich et al., 2010*; *Lim et al., 2011*; *Strait et al., 2014*), while the competition between action value representations takes place in SEF and DLPFC (*Wallis and Miller, 2003*; *Kim et al., 2008*). The selection of action value signals in turn can influence the neural activity in primarily motor-related areas, such as FEF and SC that encode the final commitment to a particular course of action (*Schall et al., 2002*; *Brown et al., 2008*; *Thura and Cisek, 2014*).

It has been suggested that different neural and functional architectures underlie different types of value-based decisions making (*Hunt et al., 2013*). In contrast, our sequential decision model predicts that decisions are always made using the same decision architecture: an initial goods-based selection, followed by action-based selection, because both stages are necessary and not functionally interchangeable. Nevertheless, the relative importance of each selection stages likely depends on the behavioral context. To understand strictly economic behavior (such as savings behavior or consumption of goods) the goods selection is the more important step. Preferred options can be selected without knowledge about the action necessary to indicate the chosen option (*Gold and Shadlen, 2003*; *Bennur and Gold, 2011*; *Grabenhorst et al., 2012*; *Cai and Padoa-Schioppa, 2014*). In such situations, there is no evidence of ongoing competition between potential actions (*Cai and Padoa-Schioppa, 2012*, *2014*). Obtaining a desired good is typically considered to be a trivial act in a well-functioning market (*Padoa-Schioppa, 2011*). On the other hand during perceptual or rule-based decision making, the action selection is the most important step in the decision

process, because only one type of good can be achieved by engaging in the task. An example are competitive games such as chess, were the goal (checkmate) is clear and implicitly chosen when a player starts a game, but were the player still has to find the most appropriate actions to achieve this goal. This implies that within different behavioral contexts, different elements of the decision circuit become critical. Altogether, we think that actual behavior under a wide range of different conditions is best understood by a model that respects that behavioral choices are the result of two independent and functionally different selection mechanisms.

## Materials and methods

Two rhesus monkeys (both male; monkey A: 7.5 kg, monkey I: 7.2 kg) were trained to perform the tasks used in this study. This study was performed in strict accordance with the recommendations in the Guide for the Care and Use of Laboratory Animals of the National Institutes of Health. All the animals were handled according to approved institutional animal care and use committee (IACUC) protocols (PR13A337) of Johns Hopkins University.

### Behavioral task

In the gambling task, the monkeys had to make saccades to peripheral targets that were associated with different amounts of reward (*Figure 2A*). The targets were squares of various colors, 2.25×2.25° in size. They were always presented 10° away from the central fixation point at a 45, 135, 225, or 315° angle. There were seven different gamble targets (*Figure 2B*), each consisting of two colors corresponding to the two possible reward amounts. The portion the target filled with each color corresponded to the probability of receiving the corresponding reward amount. Four different colors indicated four different reward amounts (increasing from 1, 3, 5 to 9 units of water, where 1 unit equaled 30 μl). The minimum reward amount for the gamble option was always 1 unit of water (indicated by cyan), while the maximum reward amount ranged from 3 (red), 5 (blue) to 9 units (green), with three different probabilities of receiving the maximum (20, 40, and 80%). This resulted in a set of gambles, whose expected value on the diagonal axis was identical, as shown in the matrix (*Figure 2B*).

The task consisted of two types of trials - choice trials and no-choice trials. All the trials started with the appearance of a fixation point at the center of the screen (*Figure 2B*), on which the monkeys were required to fixate for 500–1000 ms. In choice trials, two targets appeared on two randomly chosen locations across the four quadrants. Simultaneously, the fixation point disappeared and within 1000 ms the monkeys had to choose between the gambles by making a saccade toward one of the targets. Following the choice, the non-chosen target disappeared from the screen. The monkeys were required to keep fixating on the chosen target for 500–600 ms, after which the target changed color. The two-colored square then changed into a single-colored square associated with the final reward amount. This indicated the result of the gamble to the monkeys. The monkeys were required to continue to fixate on the target for another 300 ms until the reward was delivered. Each gamble option was paired with all other six gamble options. This resulted in 21 different combinations of options that were offered in choice trials. The sequence of events in no-choice trials was the same as in choice trials except that only one target was presented. In those trials, the monkeys were forced to make a saccade to the given target. All seven gamble options were presented during no-choice trials.

We presented no-choice and choice trials mixed together in blocks of 28 trials that consisted of 21 choice trials and 7 no-choice trials. Within a block, the order of trials was randomized. The locations of the targets in each trial were also randomized, which prevented the monkeys from preparing a movement toward a certain direction before the target appearance.

For reward delivery, we used an in-house built fluid delivery system. The system was based on two syringe pumps connected to a fluid container. A piston in the middle of the two syringes was connected with the plunger of each syringe. The movement of the piston in one direction pressed the plunger of one syringe inward and ejected fluid. At the same time, it pulled the plunger of the other syringe outward and sucked fluid into the syringe from the fluid container. The position of the piston was controlled by a stepper motor. In this way, the size of the piston movement controlled the amount of fluid that was ejected out of one of the syringes. The accuracy of the fluid amount delivery was high across the entire range of fluid amounts used in the experiment, because we used

relatively small syringes (10 ml). Importantly, it was also constant across the duration of the experiment, unlike conventional gravity-based solenoid systems.

## Estimation of subjective value of gamble options

We used Maximum Likelihood Difference Scaling (MLDS) (*Maloney and Yang, 2003*; *Kingdom and Prins, 2010*) to estimate the subjective value of different targets. The algorithm is an optimization algorithm which gives the best estimation of the subjective value and internal noise based on the maximum across-trial likelihood, which is defined as:

$$L\left(\psi(1), \psi(2), \dots \psi(N), \sigma_d | r\right) = \sum_{k=1}^{T} log_e p\left(r_k | D_k; \psi(1), \psi(2), \dots \psi(N), \sigma_d\right) \tag{1}$$

where $\psi(i)$ are the subjective value for all the targets, $\sigma_d$ is the internal noise, $r_k$ is the response (chosen:1 or non-chosen:0) on the kth trial, and $D_k$ is the estimated subjective value difference between two targets in the kth trial given the set of subjective value and internal noise,$r$ the full set of responses across all trials and $T$ the number of trials. We performed the MLDS using the Matlab based toolbox 'Palamedes' developed by Prins and Kindom (*Kingdom and Prins, 2010*).

## Neurophysiological methods and data analysis

After training, we placed a hexagonal chamber (29 mm in diameter) centered over the midline, 28 mm (monkey A) and 27 mm (monkey I) anterior of the interaural line. During each recording session, single units were recorded using 1–4 tungsten microelectrodes with an impedance of 2–4 MΩs (Frederick Haer, Bowdoinham, ME). The microelectrodes were advanced, using a self-built microdrive system. Data were collected using the PLEXON system (Plexon, Inc., Dallas, TX). Up to four template spikes were identified using principal component analysis. The time stamps and local field potential were then collected at a sampling rate of 1,000 Hz. Data were subsequently analyzed offline to ensure only single units were included in consequent analyses. We used custom software written in Matlab (Mathworks, Natick, MA), which are available at the following GitHub respository: https://github.com/XMoChen/Sequential-good-and-action-selection-during-decision-making.

## Recording location

To determine the location of the SEF, we obtained magnetic resonance images (MRI) for monkey A and monkey I. A 3-D model of the brain was constructed using MIPAV (BIRSS, NIH) and custom Matlab codes. As an anatomical landmark, we used the location of the branch of the arcuate sulcus. The locations of neural recording sites are shown in (*Figure 2—figure supplement 1*). In both monkeys, we found neurons during the saccade preparation period in the region from 0 to 11 mm anterior to the genu of the arcuate branch and within 5 mm to 2 mm of the longitudinal fissure. We designated these neurons as belonging to the SEF, consistant with previous studies from our lab and existing literature (*Tehovnik et al., 2000*; *So and Stuphorn, 2010*).

## Task-related neurons

We used several criteria to determine whether a neuron was task related. To test whether a neuron was active while the monkey generated saccades to the targets, we analyzed the neural activity in the time period between target onset to saccade initiation. We performed a permutation test on the spike rate in 50 ms intervals throughout the saccade preparation time period (150–0 ms before saccade onset or 50–200 ms after target onset) to compare against the baseline period (200–150ms prior to target onset). If p value was ≤0.05 for any of the intervals, the cell was determined to have activity significantly different form baseline. Out of 516 neurons, 353 were classified as task-related using these criteria.

Furthermore, we used a more stringent way to define the task related neuron by fitting a family of regression models to the neural activity and determining the best-fitting model (*So and Stuphorn, 2010*).

The influence of value (V) on neuronal activity was described using a sigmoid function

$$f(V) = \frac{b_1}{1 + e^{-s(V-t)}} \tag{2}$$

where $b_1$ is the weight coefficient, s (s $\in (0, 1)$) is the steepness, and t (t $\in (0, 1)$) is the threshold value. Often, the influence of expected value on neuronal activity is described using a linear function. However, SEF neurons are better described using a sigmoid function. The reasons for this are two-fold: 1) Many SEF neurons actually had a 'curved' relationship with increasing value (*So and Stuphorn, 2010*). 2) The more important reason is that a substantial number of value-related SEF neurons showed floor or ceiling effects, that is, they showed no modulation for value increases in a certain range, but started to indicate value above or below a certain threshold. In addition, the sigmoid function is flexible enough to easily approximate a linear value coding. In *Equation 2*, by setting t=0.5, b1>1, the relatively linear part of the sigmoid function can be used for value coding. Thus, the sigmoid function is flexible enough to fit the behavior of a large number of neurons with monotonically increasing or decreasing activity for varying value (including linearly related ones).

The influence of saccade direction (D) on neuronal activity was described using a circular Gaussian function

$$g(D) = b_2 \times e^{\{w \times [\cos(D-p)]^{-1}\}} \tag{3}$$

where $b_2$ is the weight coefficient, *w* (w $\in (0, 4\,\pi\,]$) is the turning width, *p* (p $\in [0, 2\,\pi\,]$) is the PD of the neuron.

The interaction of value and direction was described using the product of $f(V)$ and $g(D)$

$$h(V, D) = f(V) \times g(D) = b_3 \times \frac{1}{1 + e^{-s(V-t)}} \times e^{\{w \times [\cos(D-p)]^{-1}\}} \tag{4}$$

where $b_3$ is the weight coefficient.

For each neuron, we fitted the average neuronal activity before saccade (50ms before saccade onset to 20 ms after saccade onset) on each no-choice trial with all possible linear combinations of the three terms $f(V)$, $g(D)$, $h(V, D)$ as well as with a simple constant model ($b_0$). We identified the best fitting model for each neuron by finding the model with the minimum Bayesian information criterion (*Burnham and Anderson, 2002*; *Busemeyer and Diederich, 2010*)

$$BIC = n \times \log\left(\frac{RSS}{n}\right) + k \times \log(n) \tag{5}$$

where *n* is the number of trials (a constant in our case), and RSS is the residual sum of squares after fitting. We used a loosely defined BIC in order to include more neurons into analysis, where *k* is the number of independent variables in the equation rather than the number of parameters in the regression model. A lower numerical BIC value indicates that the model fit the data better: with a lower residual sum of squares indicating better predictive power and a larger k penalizes less parsimonious models. All neurons with lower BIC value than the baseline model containing only a constant ($b0$) were considered task related. Among the 353 task-related neurons, 128 neurons were further classified as directionally tuned and were used in the following analyses.

All neurons were tested with all 21 gamble option combinations and at least four diagonal directional combinations in which two targets where 180 degree away. One hundred and six neurons (26 from monkey A and 80 from monkey I) were tested with no less than 8 out of 12 direction combinations (4 diagonal and 4 ninety degree away in the same hemi-visual field direction combinations), and 86 neurons (6 from monkey A and 80 from monkey I) were tested with all 12 direction-combinations.

Averages of neural activity across the entire population of all 128 directionally-tuned SEF neurons were performed after the individual neurons activity was normalized by searching for the minimum and maximum activity across all choice and no-choice trial conditions and setting the minimum activity to 0 and the maximum activity to 1. The only exception is the construction of the relative action value map, were we used a slightly different definition of the zero reference point.

## Relative action value map

The normalized time-direction maps show the population activity of all directional SEF neurons based on their preferred direction relative to the chosen and non-chosen target (*Cisek and Kalaska, 2005*). For each neuron, we generated the mean firing rate separately for all 16 combinations of

choices and target configurations (choice trials: 12; no-choice trials: 4). The neuron's firing rate was normalized by setting the baseline activity (mean activity between 50 to 0 before target onset across 16 conditions) to 0 and the maximum activity across all 16 conditions to 1. Each cell's preferred direction was defined by the circular Gaussian term in the best fitting model in the BIC analysis. Population data were displayed as a 2D color plot, in which the spike density functions of each neuron were sorted along the vertical axis according to their preferred direction with respect to the location of the selected target. This resulted in a matrix in which the PD distribution within the relative action value map was unevenly sampled. The sorted matrix was therefore smoothed by linear interpolation at an angle of 7.2°. The horizontal axis showed the development of the population activity across time aligned to either target or movement onset. For all population maps, the same baseline activity and maximum activity were used for each neuron.

## Neural decoding of chosen reward option and direction

Binary linear classification was performed using Matlab toolboxes and custom code. The analysis was performed on neural activity 200 ms before till 20 ms after movement onset at a 1 ms time resolution. For each neuron, we used the neural activity in those choice trials in which the monkey chose a particular value or direction to train the classifier (around half of the trials for direction, and different numbers of trials for different values depending on the monkeys' choice behavior). We then used the classifier to predict either the direction or the value of the chosen target for each trial. When predicting the chosen direction, for example, there are two target locations in a choice trial. We used the neural activity in all choice trials when the monkey chose either one of the target locations to train the classifier, and then used the optimized classifier to predict the chosen saccade direction based on the observed neural activity in a particular trial. The overall classification accuracy was calculated by averaging across all trials for each neuron. We used a permutation test, in which we shuffled the chosen and non-chosen target value and direction, to test whether the classification accuracy was significant (1000 shuffle; $p \leq 0.05$).

## Mutual information

In order to compare the relative strength of the relationship between neural activity and saccade value and direction, we calculated separately for each neuron the mutual information between neural activity and chosen and non-chosen value or direction, respectively. To capture the dynamics of value and direction encoding, we performed the calculation repeatedly for consecutive time periods during saccade preparation using spike density at a 1 ms time resolution.

To reduce the bias in estimating the mutual information and let the estimated information comparable between trial conditions, we discretized the neural activities in the same way. During no-choice trials, we sampled the space of possible values and directions evenly, in contrast to choice trials were the values and direction depended on the monkey's preferences. We assumed that the neural activity in no-choice trials allowed us to capture how this neuron encoded value and direction information and we used neural activity in no-choice trial to determine the bins for neural activity for all trial conditions. We set the number of bins for neural activity ($N_F$) as four. At each particular time window, we collected the mean neural firing rates (F) from every no-choice trial and divided them into four bins so that each bin held equal number of no-choice trials. We then got $Q_1, Q_2$ and $Q_3$ as the boundaries for 4 quartiles. For all trial conditions, at the same time window, neural activity below $Q_1$ was classified as $F_1$, between $Q_1$ and $Q_2$ as $F_2$, between $Q_2$ and $Q_3$ as $F_3$, and finally, neural activity above $Q_3$ was classified as $F_4$.

The mutual information between neural activity F and the variable X, which can be either chosen or non-chosen value or chosen or no-chosen direction in our case, was approximated by the following:

$$I(F,X) = \left( \sum_{i=1}^{N_F} \sum_{j=1}^{N_X} \frac{M_{ij}}{M} \log \frac{M_{ij} M}{M_i \cdot M_{\cdot j}} \right) - Bias \qquad (6)$$

here $M_{ij}$ is the number of trials having both $F_i$ and $X_j$; is the number of trials having $F_i$, and $M_{\cdot j}$ is the number of trials having $X_j$. M is the number of total trials. As mentioned before, we set $N_F$, the number of distinct states of neural activity, to four. In the case of direction, we set $N_x$, the number of distinct states of the relevant variable, to four, because we tested four different saccade directions.

In the case of value, we tested seven different values. However, distinguishing seven different value levels would have resulted in different maximum amounts of mutual information for the two variables (direction: 2.00 bits; value: 2.81 bits). This would have led to an overestimation of value information relative to directional information. In order to make the value and direction information estimations directly comparable, we set $N_x$ for value to four as well. In grouping the seven different values into four bins, we followed the same binning procedure as we did for the neural activities. The chosen values were divided into four quartiles so that each bin held an equal number of no-choice trials. We computed a first approximation of the bias as follows:

$$Bias = \frac{1}{2Mlog2}(U_{FX} - U_F - U_X + 1) \tag{7}$$

where $U_{FX}$ is the number of nonzero $M_{ij}$'s for all i and j, $U_F$ is the number of nonzero $M_{\cdot i}$ for all i, and $U_X$ is the number of nonzero $M_{i\cdot}$ for all j. This procedure followed the approach described in (*Ito and Doya, 2009*).

Finally, we performed a Bootstrap procedure to test whether the amount of mutual information was significant, and to further reduce any remaining bias. We generated a random set of $F_i$ and $X_j$ pairs, by permuting both F and X arrays, respectively. We calculated the mutual information between F and X, using the same method described above, and repeated this process for 100 times. The mean of the mutual information obtained from these 100 random processes represented remaining bias and was subtracted from $I(F, X)$. To test whether the final estimated mutual information was significant (p≤0.05), we compared it with the sixth highest information obtained from the 100 random processes. If it was non-significant, we set the mutual information to zero. The bias reductions sometimes lead to negative estimates of mutual information. In that case, we also set the final estimated information to be zero.

To determine when the SEF population carried different amounts of information about the chosen and non-chosen direction or value, we compared the information about the chosen and non-chosen option across all neurons in each time bin using a paired t-test. We defined the onset of differences in information as the first time bin in which p-values were less than 0.05 for 10 or more consecutive time bins.

## Regression analysis

A linear regression was used to determine the temporally evolving contribution of the chosen and non-chosen target to the neural firing rate in choice trials. First, for each neuron, we calculated the mean firing rate on no-choice trials for each direction ($S_{no-choice}(D,t)$ or value ($S_{no-choice}(V,t)$) for sequential time steps in the trial, using a sliding time window with 20 ms width and 10 ms step size. Then, in the regression analysis, the contribution of the chosen and non-chosen directions was described as:

$$S_{choice}(t) = b_1 S_{no-choice}(D_{chosen}, t) + b_2 S_{no-chocie}(D_{nonchosen}, t) \tag{8}$$

The contribution of the chosen and non-chosen values was described as:

$$S_{choice}(t) = b_1 S_{no-choice}(V_{chosen}, t) + b_2 S_{no-choice}(V_{nonchosen}, t) \tag{9}$$

The data were fitted with a linear least-square fitting routine implemented in Matlab (The Math Works, Natick, MA).

To determine when the SEF population showed a significant (p≤0.05) difference in the influence of the chosen and non-chosen regression coefficients for direction and value, we performed paired t-tests for each time bin. We defined the onset of differences in the strength of coefficients as the first time bin in which p-values were less than 0.05 for 3 or more consecutive time bins.

## State-space analysis

Population activity can be represented within the state space framework (*Yu et al., 2009*; *Shenoy et al., 2011*). In this framework, the state of activity of all *n* recorded neurons (i.e. the activity distribution) is represented by a vector in an n-dimensional state space. The successive vectors during a trial form a trajectory in state space that describes the development of the neural activity.

Our state-space analysis follows generally the one described in *Mante et al. (2013)*. The main difference is that we did not perform a principal component analysis to reduce the dimensionality of the state space.

To construct population responses, we first computed the average activity of all recorded neurons in both monkeys for each trial condition. Then, we combined the 128 average activity values to a 128-dimensional vector array representing the population activity trajectory in state space for each trial condition. Next, we used linear regression to identify dimensions in state space containing task related variance. For the z-scored responses of neuron $i$ at time t, we have:

$$r_{i,t}\big(k\big) = \beta_{i,t}(1)chosen\_direction_{left\_right}(k) + \beta_{i,t}(2)chosen\_direction_{up\_down}(k) +$$
$$\beta_{i,t}(3)chosen\_value(k) + \beta_{i,t}(4) + \beta_{i,t}(4)nonchosen\_direction_{left\_right}(k) +$$
$$\beta_{i,t}(5)nonchosen\_direction_{up\_down}(k) + \beta_{i,t}(6)nonchosen\_value(k) + \beta_{i,t}(7)$$

$$(10)$$

where $r_{i,t}(\kappa) = \beta_{i,t}$ is the z-scored response of neuron $i$ at time $t$ on trial $\kappa$, $chosen\_direction_{left\_right}(k)$ and $nonchosen\_direction_{left\_right}(k)$ is the monkeys chosen and nonchosen direction on trial k (+1: right; -1: left), $chosen\_direction_{up\_down}(k)$ and $nonchosen\_direction_{up\_down}(k)$ is the monkeys chosen and non-chosen direction on trial k (+1: right; -1: left). There are six independent variables (var) that can influence the responses of neuron $i$ in function (10). To estimate the respective regression coefficients $\beta_{i,t}(var)$, for var=1 to 6, we define, for each unit $i$, a matrix $F_i$ of size $N_{coef} \times N_{trial}$, where $N_{coef}$ is the number of regression coeffients to be estimated and $N_{trial}$ is the number of trial recorded for neuron $i$. The regression coefficients can be then estimated as:

$$\beta_{i,t} = (F_i F_i^T)^{-1} F_i r_{i,t}$$

$$(11)$$

where $\beta_{i,t}$ is a vector of length $N_{coef}$ with elements $\beta_{i,t}(var)$, v=1–6. It corresponds to the regression coefficient for task variable $var$, time $t$, and neuron $i$. For each task variable, we build a set of coefficient vectors $\beta_{v,t}$ whose entries is $\beta_{i,t}(var)$. The new vector $\beta_{var,t}$ correspond to the directions in state space along which the task variable are represented at the level of the population.

For each task variable var, we then determined the time, $t_{var}^{max}$, for which the corresponding set of vectors $\beta_{var,t}$. $\beta_{var}^{max} = \beta_{var,t_{var}^{max}}$ with $t_{var}^{max} = argmax_t||\beta_{var,t}||$. Last, we orthogonalized the axes of direction and value with QR-decomposition. The new axis $\beta_{var}^{\perp}$ span the same 'regression subspace' as the original regression vectors; however, it each explains distinct portions of the variance in the responses. Then at a specific time t, the projections of the population response on the time-independent axes are defined by:

$$p_{v,arc} = \beta_{var}^{\perp\,T} X_c$$

$$(12)$$

where $p_{var,c}$ is the set of time-series vectors over all task variable and conditions. $X_c$ is the firing rate matrix in different trial conditions with the size of $N_{unit} \times T$.

## Acknowledgements

We are grateful to S Everling, Shreesh P Mysore and JD Schall for helpful comments on the manuscript. This work was supported by the National Institutes of Health through grant 2R01NS086104 to VS.

## Additional information

### Funding

| Funder | Grant reference number | Author |
|---|---|---|
| National Institute of Neurological Disorders and Stroke | R01NS086104 | Veit Stuphorn |

The funders had no role in study design, data collection and interpretation, or the decision to submit the work for publication.

## Author contributions

XC, Conception and design, Acquisition of data, Analysis and interpretation of data, Drafting or revising the article; VS, Conception and design, Drafting or revising the article

## Ethics

Animal experimentation: This study was performed in strict accordance with the recommendations in the Guide for the Care and Use of Laboratory Animals of the National Institutes of Health. All of the animals were handled according to approved institutional animal care and use committee (IACUC) protocols (PR13A337) of Johns Hopkins University.

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
