## [Decision Letter]

Thank you for submitting your work entitled "Sequential selection of economic good and action in medial frontal cortex of macaques during value-based decisions" for peer review at *eLife*. Your submission has been favorably evaluated by Timothy Behrens (Senior editor), a Reviewing editor, and two reviewers.

Although both reviewers were supportive of the theme of your work and the experimental procedures and data, they voiced a number of concerns that need to be addressed carefully. You will find these concerns carefully numbered, and we hope you can address them all in a satisfactory way. In addition, the Reviewing editor has made the following comments:

1) It is unclear how you define 'preferred direction' and 'non-preferred direction'. This may be hidden somewhere in the text, but this must be made in a prominent place. The question is whether the preference relates to spatial position or direction (spatial preference), or whether it relates to the larger reward value obtained from movement in a particular direction (preference based on reward).

2) Only when this point has been clarified can we understand what you mean by 'action value': is this the value obtained by a specific action *irrespective* of that action being performed (the correct definition for an input to a competitive decision mechanism: an action value neuron would reflect that value irrespective of the animal choosing this action; and for each different action there would be a different pool of action value neurons that compete at the input), or is this the value obtained when the animal actually makes this action but not when it doesn't make that action (which is called chosen value, which is usually not considered an input to a competitive decision mechanism). It would be good to present these definitions clearly immediately after having defined 'preferred direction' and 'non-preferred direction'.

3) I would recommend transferring the interpretation of a difference signal in terms of inhibitory interactions between (action value?) neurons from the Results into the Discussion, which would also partly address one of the referees' issues. The results are fine as they are, but their interpretation is debatable in the absence of a visible inhibitory response.

4) In Figure 2, how did you arrive at the EV values of 1.8, 2.6 and 4.2, and how can you say that all points on the same diagonal have same EV? E.g. 3 x 0.4 = 1.2 but 5 x 0.2 = 1.0 for the lower diagonal, and the other two diagonals present the same issue.

5) I would recommend not to put any abbreviations in the figures, even if they are explained in the legends. This makes the figures less understandable, and the paper less cited. There is always a more informative way to indicate what is required.

*Reviewer #1:*

Most authors agree that economic decisions entail computing and comparing the subjective values of available options. There is also general agreement that subjective values are computed in the OFC/vmPFC, with the possible participation of other areas such as the amygdala. However, there is no general consensus of where and how exactly subjective values are compared to make a decision. Several models have been put forth in the past few years, as described in Figure 1 of this manuscript. These models are not necessarily mutually exclusive – for example, Paul Cisek who proposed the "distributed consensus" model, also pointed out that economic decisions can be made in a good-based representation (Cisek, 2012). However, even if one accepts the idea that each model has some domain of applicability, data shedding light on the relative extent of these domains are very valuable. In this study Chen and Stuphorn use a clever task design, in which monkeys choose between different gambles. Across trials, the expected values of the chosen and non-chosen gambles vary in ways that allow several insightful analyses. The animals revealed their choices through eye movements and the authors recorded and analyzed the activity of neurons in the supplementary eye fields (SEF). Contrary to most neurophysiology studies, here the trial structure did not impose a delay between the offer and the saccade targets, which provided a measure of the RTs as a function of the decision difficulty. The lack of a delay imposed particular care in the analysis of neuronal data, but the senior author is an expert of this sort of analyses and the approach(es) taken here are adequate. The main results of the study are (1) both information about value options and saccade directions are present in the SEF, (2) the decision between values is resolved significantly earlier than that between actions, and (3) in SEF there is evidence of mutual inhibition between the signals associated to different possible saccades (but this effect is not present in signals associated to different possible values). The authors conclusions are (A) in this task design decisions are made sequentially, first in goods space (selection of one value/option) and then in actions space (selection of a suitable action) and (B) SEF participates in the process of action selection, but not in the process of value/option selection. For the reasons outlined above, (A) is particularly notable because the task design did not impose or even invite monkeys to make the primary choice in goods space. In other words, this would have been a perfect situation for an action-based decision, but even in this situation the evidence is against that model. The authors thus propose a new "sequential decision" model, which differs from the good-based model in that both possible actions are computed and selected between. My view of the study is positive overall, although I have several reservations. The experiment was well designed, the data analysis is for the most part credible, and the discussion of the results is mostly appropriate. My main concerns are due to the fact that in a number of circumstances the authors do not provide sufficient information and/or their description is unclear – specific instances are indicated below. I'd like to see a revision of the paper, but if the authors can address these issues adequately, I would consider the study worth of publication.

1) The authors recorded from 516 neurons but only 128 entered the analysis. I am confused about the selection criteria. It appears that 353 cells were task-related, but only about one third of them made it into the analysis. Was this because the remaining 2/3 did not have a response field? I thought that the majority of neurons in SEF are selective for saccades in some direction. Also, the authors established the "influence" of value using the function in [Disp-formula equ2]. I am not sure where this function came from, but there are many cases in the literature in which value was encoded in a linear way, which is at odds with the present assumption. The authors should elaborate on their criteria and also test alternative forms of value encoding.

2) The evidence showing that neuronal activity is positively related to the value in the PD and negatively related to the value in the NPD is convincing. However, the authors repeatedly state that neurons encode the *difference* in action values. This is a much more stringent statement and there is no evidence to support it. For example, the influence of NPD could be through mechanisms of divisive normalization akin to those observed in area LIP (Louie et al.). If the value modulation was really according to the value difference, one would expect the two regression coefficients shown in Table 1 to be similar in absolute value. In contrast, the absolute value of the regression coefficient for PD is roughly twice that for NPD. Incidentally, the authors should provide a confidence interval for these coefficients.

3) Throughout the analysis, the authors treat non-choice trials separately from choice trials. But it is not clear that this is the most appropriate way to think about the data. In many studies, non-choice trials are considered forced choices in which simply one of the two values is zero, and all trials are pooled. This is relevant to the issue discussed in the previous point because if the value modulations was according to the value difference, forced choices in the PD should present the highest activity, and forced choices in the NPD should present the lowest activity (all for given chosen value). I don't see evidence of this in Figure 4, comparing panes A and B, although I am not sure how exactly the activity was normalized for these plots (please describe). Also, it would be good to see the equivalent of Figure 4 for no-choice, NPD trials.

4) To reiterate the point, unless the authors can provide clear evidence for modulation based on value difference as distinguished from other context dependent value modulation, they should remove the phrase "value difference" completely from the manuscript.

5) Figure 4 presume that all the data shown in the figure are conditioned on the animal choosing the target in the PD (please clarify!). If so, the number of trials participating to the yellow line in Figure 4 should be fewer than those participating to the red line in the same panel. Similarly in Figure 4, there should be few/more trials in the red/yellow line. This difference in number of trials, which should be very substantial, should translate into a difference in the SEM traces shown in the figure, and I am puzzled by the fact that I don't see any of that.

6) Subsection “SEF neurons reflects value difference between offered gamble options”: I am not convinced by the argument that mutual inhibition implies that the neuronal population participates in the decision (between actions), while lack of such mutual inhibition (a.k.a., menu invariance) implies that this neuronal population does not participate in the decision (between values/options). For example, context effects of the sort described here are relatively minor in area MT, even though neurons in MT clearly participate in, or are upstream of, the decision. Similarly, "offer value" cells in OFC, which are thought to provide the input for value-based decisions, don't show such effects of mutual inhibition. So this is a tricky argument, even though I agree that neurons in SEF are likely downstream of the decision between values while they participate in the decision between actions.

*Reviewer #2:*

The authors report on a study in which they trained monkeys to make choices among pairs of gambles that differed in reward magnitude and probability. Monkeys chose well, preferring options with larger expected values. They examined neural activity in the SEF while monkeys carried out the task and found that the representation of the chosen option occurred about 120 ms before saccade onset and then decreased before the representation of the chosen direction about 40 ms before saccade onset. They further found that the response to reward magnitude did not scale responses in the unchosen direction; the response in the chosen direction did depend on the target that was not chosen. Further, there was an increase in activity for neurons that represented both the chosen and unchosen options before the activity diverged and began to represent almost only the chosen option. They interpret this effect in terms of a competition model of decision making.

This paper addresses an important question about the contribution of the SEF to choices among gambles. The study was carefully carried out and the results are clearly and thoroughly presented. The task and the analytical approach to the task clarify several points about the representation of choices and actions to obtain those choices. I have several comments, however which, if addressed, could further clarify the results.

1) My two main comments have to do with the interpretation of the data relative to the models. First, how do the authors eliminate the possibility that they are observing a read-out of a choice process (possibly mutual-inhibition) that is actually occurring elsewhere? The fact that activity for both options begins to increase and then separate does not clear rule out this possibility.

2) If I had to choose one of the models in Figure 1 that was most consistent with the data, I would probably choose model B. Can the authors address more directly why their data is more consistent with model D and not model B? Is the evidence for a recurrent choice process of mutual inhibition the presence of activity representing both options for a period before they diverge? What other mechanisms could give rise to this? Is it really just mutual inhibition? Is there evidence for a representation of both choice options for a period before they diverge? An analysis similar to the one shown in Figure 6 for options instead of directions could clarify this.

[Editors' note: further revisions were requested prior to acceptance, as described below.]

Thank you for resubmitting your work entitled "Sequential selection of economic good and action in medial frontal cortex of macaques during value-based decisions" for further consideration at *eLife*. Your revised article has been favorably evaluated by Timothy Behrens (Senior editor), a Reviewing editor, and two reviewers. The manuscript has been improved but there are some remaining issues that need to be addressed before acceptance, as outlined below:

1) Both referees have some remaining issues that we would like you to address carefully.

2) The Reviewing editor is unhappy about your reply concerning action value (your point #2 and the text 'The SEF neurons encode therefore the action value of saccades to the PD target'): of the three types of action value that you mention, only the first type is real action value. The other two types are a combination of action values (your type 2) and straight chosen value (your type 3, which is the opposite of action value, a decision output variable). To avoid confusion in the community, please accept the only valid definition of action value, which comes out of machine learning: the value of an action that is *independent* of the action being chosen (a decision input variable). To me it looks like you did not find coding of action value of your type 1. Please rephrase your description of the type of value being coded.

*Reviewer #1:*

My concerns were addressed for the most part. However there remain a few relatively minor issues. The numeration below refers to the points raised in my first review.

1) How many neurons are "task-related"? In the first paragraph of the subsection “Task-related neurons” the authors state 353; in the fifth paragraph they state 362. Also, in the rebuttal, they say that the results reported in the paper, obtained with the smaller population of 128 neurons that pass the BIC criterion, are valid also for the larger population of 353/362 task-related cells. This fact should be reported in the paper (the additional figure could be included as supplementary material).

In the rebuttal the authors explain why they fitted responses with a sigmoid. The two reasons – (1) there are cells with "curved" tuning and (2) there are cells with floor/ceiling effects – seem to me one and the same. In any case, the authors should clarify this point in the Methods, not just with this reviewer.

3) Pooling forced choice trials with other trials. Here the authors did not really address my point. The question is this: If instead of separating forced choices for other trials the authors had pooled all trials, as was done in many other studies, would the results presented here be affected in any significant way?

*Reviewer #2:*

It would be useful if the authors provided additional discussion, in the manuscript, on the predictions they believe each of the models make, and how their data is consistent/inconsistent with each of the models. They have provided this information in the replies to the reviewers, but it needs to be in the manuscript. Otherwise, I have no further comments.

---

## [Author Response]

*Although both reviewers were supportive of the theme of your work and the experimental procedures and data, they voiced a number of concerns that need to be addressed carefully. You will find these concerns carefully numbered, and we hope you can address them all in a satisfactory way. In addition, the Reviewing editor has made the following comments:*

*1) It is unclear how you define 'preferred direction' and 'non-preferred direction'. This may be hidden somewhere in the text, but this must be made in a prominent place. The question is whether the preference relates to spatial position or direction (spatial preference), or whether it relates to the larger reward value obtained from movement in a particular direction (preference based on reward).*

We thank the editor for pointing this out. We define “preferred direction” (PD) as the saccade direction for which a neuron is maximally active, irrespective of reward value obtained by the saccade. PD therefore refers to spatial preference. This is illustrated in the revised Figure 4, which shows the activity of the SEF neurons during no-choice trials, i.e. when only one target appears on the screen. While the SEF neurons are strongly active for saccades into the preferred direction and show value-related modulations (Figure 4), they are not active for saccades into the non-preferred direction, even when this leads to large reward amounts (Figure 4). We agree with the reviewer that the definition of ‘preferred direction’ (PD) is important and should be in a prominent place. In the revised manuscript, we added an explanation in the Results section, when we first refer to PD (subsection “SEF neurons reflect the value of both choice options in an opposing way”). In addition, we explain how PD is estimated in more detail in the Method part of our manuscript. We used a nonlinear regression to estimate value modulation and direction modulation simultaneously for each neuron. Therefore, the estimation of preferred direction is based on the residual neuronal activity variation, after the influence of reward was taken into account. In addition, in our experiment design, all the directions were assigned with all possible reward. Thus, the reward amount and direction was dissociated from each other (across different trials).

*2) Only when this point has been clarified can we understand what you mean by 'action value': is this the value obtained by a specific action* irrespective *of that action being performed (the correct definition for an input to a competitive decision mechanism: an action value neuron would reflect that value irrespective of the animal choosing this action; and for each different action there would be a different pool of action value neurons that compete at the input), or is this the value obtained when the animal actually makes this action but not when it doesn't make that action (which is called chosen value, which is usually not considered an input to a competitive decision mechanism). It would be good to present these definitions clearly immediately after having defined 'preferred direction' and 'non-preferred direction'.*

We thank the editor for asking us to clarify, what we mean by action value. We would define it as neural signals that combine information that specifies a particular type of action (e.g. saccade direction and amplitude) with information about the value associated with the outcome of that action. There are a number of subtypes of action value signals that are related to the different stages of the decision process: 1) Signals that represent the value of the alternative actions irrespective of the choice (this is what the editor has in mind, the input on which a decision is based), 2) Signals that represent the essential step in decision making, namely the comparison between the action values of the various alternatives. Such signals should be positively correlated with the action value of one alternative and negatively correlated with the action value of the other alternatives, 3) Signals that indicate the value of the chosen action (the output of the decision process). We have found in SEF ‘action value’ signals of the 2^nd^ and 3^rd^ type, i.e. comparative action value signals that develop into chosen action value signals. Please see our clarification in the new revision (subsection “SEF neurons reflect the value of both choice options in an opposing way”).

*3) I would recommend transferring the interpretation of a difference signal in terms of inhibitory interactions between (action value?) neurons from the Results into the Discussion, which would also partly address one of the referees' issues. The results are fine as they are, but their interpretation is debatable in the absence of a visible inhibitory response.*

We feel that we show convincingly that the activity of the SEF population is positively correlated with the value of the PD and negatively correlated with the value of the NPD target (Figure 4). We think it is very hard to explain this finding without presuming some form of inhibitory interaction between the alternatives at some stage in the decision process. However, we agree with the editor and the reviewers that there are a number of different possibilities regarding the exact nature of the inhibitory interactions and the stage in the decision process at which they take place. They could simply reflect activity in SEF that is correlated with the decision process that takes place in other brain areas, or they could reflect inhibitory processes within SEF. Likewise, local inhibitory processes could take the form of specific mutual inhibition between SEF neurons with different PD, or global inhibition, or a mixture of both. Accordingly, we followed the suggestion of the editor and shifted text describing the interpretation of the result to the Discussion section. We left some references to inhibitory interactions in the Results section to motivate particular analyses. Without it, we felt the paper would be harder to follow.

*4) In Figure 2, how did you arrive at the EV values of 1.8, 2.6 and 4.2, and how can you say that all points on the same diagonal have same EV? E.g. 3 x 0.4 = 1.2 but 5 x 0.2 = 1.0 for the lower diagonal, and the other two diagonals present the same issue.*

We apologize for the confusion. For each target, the red, blue and green colors denote the maximum reward amounts the animals can receive, while the cyan color denotes the minimum reward amount the animal can receive. The minimum reward amount is always 1 (and not 0 as the editor quite naturally assumed). For example, the expected value of the red/cyan targets is calculated as 3 (maximum reward) *0.4 (maximum reward probability) +1 (minimum reward) *0.6 (minimum reward probability), which adds up to an EV value of 1.8. In the revision, we added some description to the legend of Figure 1 to better explain this fact.

*5) I would recommend not to put any abbreviations in the figures, even if they are explained in the legends. This makes the figures less understandable, and the paper less cited. There is always a more informative way to indicate what is required.*

We thank the editor for the helpful suggestion. We changed all the abbreviations in the figures into their spelled out version when there is enough space.

Reviewer #1:

*[…] I'd like to see a revision of the paper, but if the authors can address these issues adequately, I would consider the study worth of publication.*

*1) The authors recorded from 516 neurons but only 128 entered the analysis. I am confused about the selection criteria. It appears that 353 cells were task-related, but only about one third of them made it into the analysis. Was this because the remaining 2/3 did not have a response field? I thought that the majority of neurons in SEF are selective for saccades in some direction.*

We thank the reviewer for pointing this out. It is indeed true that earlier recording studies in SEF have reported that large majorities of SEF neurons showed saccade direction-related selectivity (Schlag and Schlag-Rey, 1987; Schall, 1991; Schlag et al., 1992; Chen and Wise, 1995a, b; Hanes et al., 1995; Olson and Gettner, 1995; Russo and Bruce, 1996, 2000; Isoda and Tanji, 2002; Moorman and Olson, 2007; Stuphorn et al., 2010). However, most of these studies were performed with only one electrode at a time, and often with the capability to record only the action potentials of one neuron at a time (this was simply the technical standard in the 80-90ies of the last century). This required the use of strong selection criteria for recording neurons. Typically, a researcher would search for some time within a population of neurons for cells, whose activity patterns are of interest. Clearly, such a recording scheme introduces a strong recording bias. Therefore, the numbers concerning the frequency of neurons with particular functional criteria in older papers has to be taken cautiously. In contrast, in our study we used multiple electrodes (2-4) and we were able to identify an unlimited number of individual spike waveforms, since we recorded the analog voltage recording at each electrode (the current technical standard). We also started to record only after we had found at least one ‘interesting’ neuron on an electrode, but nevertheless we recorded a much larger number of surrounding SEF neurons. Accordingly, our sample of recorded SEF neurons includes a much larger number of ‘randomly’ sampled neurons. It is therefore maybe not too much of a surprise that we found a large number of neurons that did not seem to be engaged in the task.

Thus, the topic of our study is not SEF as a whole or even a large majority of SEF neurons, but rather the attributes of a particular subclass of SEF neurons that had characteristics that were of interest to us. We defined task-related very generously as any neuron that modulated its activity relative to baseline (the time immediately before trial start, i.e., fixation cue onset). We were interested in neurons with saccade-direction modulation and selected neurons according to the Bayesian information criterion (BIC) as described in the Methods (subsection “Task-related neurons“). The BIC is a very strict criterion, and therefore a large number of neurons with weak direction or value modulation were not included in the analysis. However, the population results with 353 neurons are similar to the result with only 128 neurons, but just overall relatively weaker. For example, Figure 8 shows the same result of Figure 4 in the main text, but across all 353 task-related neurons.

Author response image 1.SEF neurons represent the difference in action value of saccades in the preferred and non-preferred direction.The neural activity of 353 directionally tuned SEF neurons was normalized and compared across trials with different values of the preferred direction or non-preferred direction target. (**A–B**) The neural activity in no-choice trials when the saccade in the preferred direction (**A**) and non-preferred direction (**B**) was performed. (****C–D****) The neural activity in choice trials. To visualize the contrasting effect of the preferred and non-preferred target on neural activity, the value of one of the targets was held constant, while the value of the other target was varied he value of the preferred target varied, while the value of the non-preferred target was held at a medium value. Alternatively, the value of the non-preferred target varied, while the value of target was held at a medium value. The color of the spike density histograms indicates the target value [high value=6-7 units (red line); medium value=3-5 units (orange line); low value=1-2 units (yellow line)]. (****E–H****) The regression analysis corresponding to (A–D). A t-test was used to determine whether the regression coefficients were significantly different from 0.**DOI:**
http://dx.doi.org/10.7554/eLife.09418.021

In sum, we are confident that our selection criteria did not distort our findings regarding the activity of SEF neurons in the gamble task.

*Also, the authors established the "influence" of value using the function in [Disp-formula equ2]. I am not sure where this function came from, but there are many cases in the literature in which value was encoded in a linear way, which is at odds with the present assumption. The authors should elaborate on their criteria and also test alternative forms of value encoding.*

We first introduced the use of the sigmoid function in an earlier publication (So and Stuphorn, 2010), in which we described SEF neurons that encode both saccade direction and expected value (i.e., action value). The reason we used a sigmoid instead of a linear function is because it resulted in a better fit and in a larger number of neurons in which we could describe the relationship with value. The reasons for this are two-fold: 1) Many SEF neurons actually had a ‘curved’ relationship with increasing value. An example of this is shown in Figure 9 from So and Stuphorn (2010). 2) The more important reason is that a substantial number of value-related SEF neurons showed floor or ceiling effects, that is, they showed no modulation for value increases in a certain range, but started to indicate value above or below a certain threshold. In general, we agree with the reviewer that many neurons encode value in a relative linear way. However, the sigmoid function is flexible enough to easily approximate a linear value coding. For example, in [Disp-formula equ2], by setting t=0.5, b1>1, the relatively linear part of the sigmoid function then can be used for value coding. Thus, the sigmoid function is flexible enough to fit the behavior of a large number of neurons with monotonically increasing or decreasing activity for varying value (including linearly related ones). The main concern of the reviewer, namely that we might have missed such neurons seems therefore unwarranted. Of course, it might be argued that the additional parameters of the sigmoid function are not necessary to capture the main trend and that there might be a danger of over-fitting. However, this seems unlikely, given the fact that the BIC criterion was rather selective (see #1 above). In any case, we used the multiple regression analysis only for classification purposes. None of the main analysis depends on using the sigmoid function.

*2) The evidence showing that neuronal activity is positively related to the value in the PD and negatively related to the value in the NPD is convincing. However, the authors repeatedly state that neurons encode the difference in action values. This is a much more stringent statement and there is no evidence to support it. For example, the influence of NPD could be through mechanisms of divisive normalization akin to those observed in area LIP (Louie et al.). If the value modulation was really according to the value difference, one would expect the two regression coefficients shown in Table 1 to be similar in absolute value. In contrast, the absolute value of the regression coefficient for PD is roughly twice that for NPD.*

We apologize for the misunderstanding here. We made multiple changes in the revised version of the paper to clarify this issue (Results).

We fully agree with the reviewer that the SEF neurons do not encode the difference in action values in the strict mathematical sense. Our reference to the ‘difference in action value’ was meant as a description of the fact that the SEF neurons represent the action value strength of saccades in the preferred direction relative to the strength of saccades in the non-preferred direction. Thus, they signal a relative, not absolute, action value signal. The reviewer is of course quite right in pointing out that our terminology is misleading. Clearly, the effect of the negative modulation by alternative targets is not as strong as the effect of the positive modulation by the preferred target. This is in fact an interesting finding and we now note it in the revised manuscript (subsection “SEF neurons reflect the value of both choice options in an opposing way”). Nevertheless, despite the fact that the effect of PD and NPD is not exactly equal at the level of SEF neurons, the neural activity pattern implies that somewhere in the neural circuit leading from the target detection and evaluation to the SEF activity we record there is at least one (possible multiple) steps in which the neurons representing the competing targets exert an inhibitory effect on each other, which embodies a competitive interaction. Of course, based on the existing data we cannot say if this step occurs: 1) upstream of the SEF, 2) locally within the SEF, or 3) whether it occurs at multiple levels within the decision network. The demonstration of direct local inhibition between SEF neurons would require experiments such as the ones performed by the Schlag’s in FEF and SC (Schlag-Rey et al., 1992; Schlag et al., 1998) in which they recorded activity in some oculomotor neuron and demonstrated that activation of a different part of FEF or SC led to decreased activity in the neuron, or optogenetic stimulation experiments (Nassi et al., 2015). However, given the knowledge that inhibition plays an important role in many other cortical areas, including FEF and LIP (Schlag et al., 1998; Falkner et al., 2010), we think the suggestion that this might also be the case in SEF is not far-fetched and certainly would fit with our results.

Likewise, there are multiple possible inhibitory mechanisms within the SEF that could give rise to the activity pattern we describe. The inhibition could operate directly between SEF neurons with different PDs (some form of lateral inhibition). Alternatively, a global inhibitory network could be involved as in standard neuronal decision models (Wang, 2002), or as in divisive normalization (Nassi et al., 2015)). The exact mechanism (or mechanisms) must be of course worked out in future experiments. Nevertheless, it seems important to note that all of these mechanisms have the same functional consequence: they generate a situation, in which the incentive to choose one course of action is suppressed by the simultaneously existing incentive to choose alternative actions. Thus, the various action value signals have to compete with each other, which constitutes the essential step in decision-making.

We thank the reviewer for this comment that encouraged us to discuss the different possible interpretations of our data in more detail and to distinguish more clearly between data and interpretation.

*Incidentally, the authors should provide a confidence interval for these coefficients.*

Please see the new Table 1.

*3) Throughout the analysis, the authors treat non-choice trials separately from choice trials. But it is not clear that this is the most appropriate way to think about the data. In many studies, non-choice trials are considered forced choices in which simply one of the two values is zero, and all trials are pooled. This is relevant to the issue discussed in the previous point because if the value modulations was according to the value difference, forced choices in the PD should present the highest activity, and forced choices in the NPD should present the lowest activity (all for given chosen value). I don't see evidence of this in Figure 4, comparing panes A and B.*

We thank the reviewer for this insightful comment. In general, we concentrated in this manuscript on choice trials simply because only these trials allow us to study the decision process and the interaction of the two targets on each other. With respect to the encoding of value difference, we agree with the predictions of the reviewer. Activity on no-choice trials choices in the PD, the activity should be highest. Instead, the activity on choice trials with a medium value NPD target, was higher than on no-choice trials, when the PD target had a higher value, but lower, when it had a smaller value (compare Figure 4 with Figure 4). This shows that SEF neurons encode a relative action value signal, but not the exact mathematical difference between the absolute value amounts.

*Although I am not sure how exactly the activity was normalized for these plots (please describe).*

In the revised manuscript, we describe the normalization method in more detail (subsection “Task-related neurons”). In brief – for each trial of a particular neuron, we generate a smooth spike density function. Within this trial set, we then search for the maximum and minimum neuronal activity across all trial conditions (choice and no-choice trials) and re-scale the spike density functions, so that the maximum activity is 1 and the minimum activity is 0. Thus, the activity scale of the normalized activity is the same across all trial types and the activity can be directly compared.

*Also, it would be good to see the equivalent of Figure 4 for no-choice, NPD trials.*

Please see the new Figure 4 and new Figure 4—figure supplement 1.

*4) To reiterate the point, unless the authors can provide clear evidence for modulation based on value difference as distinguished from other context dependent value modulation, they should remove the phrase "value difference" completely from the manuscript.*

We agree with the reviewer that “value difference” is a potentially misleading term and removed it from the manuscript. In the revised version, we discuss in detail the different possible interpretations of our findings, as well as their general meaning (Discussion).

*5) Figure 4 presume that all the data shown in the figure are conditioned on the animal choosing the target in the PD (please clarify!). If so, the number of trials participating to the yellow line in Figure 4 should be fewer than those participating to the red line in the same panel. Similarly in Figure 4, there should be few/more trials in the red/yellow line. This difference in number of trials, which should be very substantial, should translate into a difference in the SEM traces shown in the figure, and I am puzzled by the fact that I don't see any of that.*

We apologize for this misunderstanding that is caused by a misleading figure labeling. We have corrected the figure legend and thank the reviewer for noticing the ambiguity. In Figure 4, the trials (different colored lines) are sorted by the value of the PD and NPD target regardless of the saccade direction that the monkey chose. The number of trials is therefore similar in all conditions, which explains that the SME traces have a similar range. We did this because we want to clearly demonstrate that the neuronal activity was influenced by both PD and NPD value (in an inverse fashion) independent of the additional complication of saccade choice. It is of course true that the SEF activity reflects not only PD and NPD value, but also chosen saccade direction. We decided therefore to analyze the SEF activity by PD and NPD value, as well as by chosen saccade direction. The results are included in the revised manuscript as Figure 4—figure supplement 2. The trials are sorted by direction (preferred: right; non-preferred: left) and value of the chosen or non-chosen target. We grouped the subjective value of the reward options into three groups (red: high value; orange: medium value; yellow: low value). The inset above the histograms indicates the location and value of the targets in the trials shown in the histograms below. The grey oval indicates the location of the preferred direction of the neuron, while the arrow indicates the chosen saccade direction. The p-values indicate the significance of a regression using all 7 individual target values without grouping. The shaded areas represent standard error of the mean (SEM).

Figure 4—figure supplement 2 shows the neural activity in no-choice trials. The color of the spike density histograms indicates the target value. As shown in Figure 4 in the revised manuscript, the neurons reflect the value of the PD target, but not of the NPD target. Figure 4—figure supplement 2 shows the neural activity in those choice trials, in which the chosen target had the highest possible value (7 units). We chose this reference point, instead of a medium value (as in Figure 4), because it allowed us to a comparison with the widest possible range of non-chosen target values (indicated by the color of the spike density histograms). Figure 4—figure supplement 2 shows the neural activity in those choice trials, in which the non-chosen target had the lowest possible value (1 unit). Again, this reference point allowed for the widest possible range of chosen target values (indicated by the color of the spike density histograms).

Consistent with Figure 4, we can observe that increasing PD target value increases neuronal activity (Figure 4—figure supplement 2 left panel; Figure 4—figure supplement 2 right panel), while increasing NPD target value decreases neuronal activity (Figure 4—figure supplement 2 left panel). This is true, whether the PD or the NPD target is chosen. However, the value of the chosen target has a stronger influence than the non-chosen one, and in the extreme case, when the PD target has the highest possible value and is chosen, the NPD value has no significant effect on the neural activity (Figure 4—figure supplement 2 right panel).

*6) Subsection “SEF neurons reflects value difference between offered gamble options”: I am not convinced by the argument that mutual inhibition implies that the neuronal population participates in the decision (between actions), while lack of such mutual inhibition (a.k.a., menu invariance) implies that this neuronal population does not participate in the decision (between values/options). For example, context effects of the sort described here are relatively minor in area MT, even though neurons in MT clearly participate in, or are upstream of, the decision. Similarly, "offer value" cells in OFC, which are thought to provide the input for value-based decisions, don't show such effects of mutual inhibition. So this is a tricky argument, even though I agree that neurons in SEF are likely downstream of the decision between values while they participate in the decision between actions.*

We thank the reviewer for helping us to clarify our argument. We agree with the reviewer that the mutual inhibition is not a necessary condition for neural signals to participate in the decision process. Our argument is slightly different. The decision process consists of: 1) a representation of the alternatives (e.g. options, actions, or perceptual states), 2) a comparison of the alternatives, and 3) a representation of the chosen alternative. All 3 types of variables are necessary, but they indicate different stages of the decision process. We observe in SEF neuronal signals that match the last stage of the economic goods selection, i.e., a chosen option signal. This is evident in the asymmetric effect of chosen option in the regression analysis (Figure 5), but also in the fact that there is almost no increase in mutual information about the non-chosen option (Figure 3). This is in contrast to action-related signals, which imply that SEF represent the comparison stage and the chosen alternative stage in the case of action selection. Altogether, we would argue this suggests that SEF receives the output of the option selection process that takes place upstream of SEF, but participates in the action selection process. It seems that this interpretation is not too far off the reviewer’s ideas. We agree that our line of reasoning is not completely decisive. In the revised paper, we therefore clarified our reasoning and made sure to indicate clearly that this is only our interpretation of the data (subsection “SEF neurons reflect the value of both choice options in an opposing way”).

Reviewer #2:

*1) My two main comments have to do with the interpretation of the data relative to the models. First, how do the authors eliminate the possibility that they are observing a read-out of a choice process (possibly mutual-inhibition) that is actually occurring elsewhere? The fact that activity for both options begins to increase and then separate does not clear rule out this possibility.*

We agree with the reviewer that our results do not rule out that the neuronal processes in SEF might reflect processes that actually occur in some other part of the brain. However, our results are also compatible with the possibility that SEF at least partly participates in the action selection. More importantly, our main results regarding the sequential nature of decision making is independent of this question. We have tried to be clear about this in the previous manuscript and have made additional modifications to be even clearer in this revised version (Discussion).

In fact, it seems likely that SEF is part of a larger network of cortical and subcortical areas (including vLPFC, dLPFC, FEF, ACC, caudate) that all participate in the value-based decision. We know that many of these regions are closely interconnected and that activity in these regions is likely correlated. On the other hand, the fact that SEF is part of a larger network and is influenced by the ongoing activity in this network does not preclude an active role of SEF in shaping the dynamic and outcome of this process. Separating and identifying the specific role of all the different nodes in this network is a complex process and will require a number of additional future experiments, but this is true for practically all neurophysiological recordings at the present time.

Our main results are independent of whether SEF actively participates in the decision process or only passively reflects it. In either case, the SEF activity allows us to describe two central elements of value-based decision making: 1) decisions are made sequentially, with option selection preceding action selection, and 2) the selection process involves some form of inhibitory competition between the neural representations of the different alternatives. We think that the first point is the most novel contribution of our work and should go some way to help understanding of value-based decision making. The second point is of course not so novel and should not be controversial. We hope we made clear in our response to Reviewer 1 that we do not wish to make claims about more specific aspects of the neural architecture underlying the competition, which also await future experiments to uncover.

*2) If I had to choose one of the models in Figure 1 that was most consistent with the data, I would probably choose model B. Can the authors address more directly why their data is more consistent with model D and not model B?*

If the pure action-selection model B were correct, the competition would only happen in the action value space. This is graphically indicated by the absence of inhibitory connections between the nodes representing the reward options (or goods). In this model B, we would expect the chosen direction information to appear simultaneously with or even slightly earlier than the chosen reward option information, since the selected action value signal contains both direction and value information. However, this prediction is contradicted by our observation that the chosen value information is present earlier than the chosen action information (Figure 3, Figure 5 and Figure 7). Thus, there is a moment during the decision-making process (100-50 ms before saccade onset), when the SEF neurons encode which option is chosen, but not yet, what saccade will be chosen. This could clearly never happen in an action selection model of decision-making. On the other hand, we found also evidence for competition between action value-encoding neurons in SEF that is spatially organized, i.e., in an action-based frame of reference (Figure 4). This fact, together with the fact that we find activity corresponding to both response option (Figure 6) clearly rules out the pure good-based model A. The time difference in the onset of chosen option and direction information also argues against strong recurrent connections from the action selection level back to the option selection level, as suggested by the distributed consensus model C. Thus, our data are most consistent with a model that predicts selection both on the option and the action representation level, but with asymmetric connections between them, so that the option selection level influences the action selection level, but not vice versa. This is the sequential model D.

*Is the evidence for a recurrent choice process of mutual inhibition the presence of activity representing both options for a period before they diverge? What other mechanisms could give rise to this? Is it really just mutual inhibition?*

The most important evidence for the presence of mutual inhibition is the fact that, as shown in Figure 4, the neuronal activity is negatively correlated with the action value in NPD targets (A) while the neuron does not response to the NPD target if it appears alone (B). Thus, the PD target value representation is modulated in negatively in proportion to the NPD target value. As shown in Figure 6, the magnitude of negative contribution and the speed with which one representation is influenced by the other is closely related to the relative strength of each representation. We hypothesize that these effects are caused by inhibitory mechanisms. As discussed in our reply to Reviewer 1, there is a number of inhibitory mechanisms that would be in line with this finding (mutual inhibition of local neuron pools across SEF, or activation of a global pool of inhibitory neurons). In the revised paper, we now discuss these different possibilities (Discussion).

*Is there evidence for a representation of both choice options for a period before they diverge? An analysis similar to the one shown in Figure 6 for options instead of directions could clarify this.*

In addition to the ‘static’ argument following from the differential effect of PD and NPD value on SEF activity, there is also a ‘dynamic’ argument in the sense that a process of mutual inhibition should lead from a state of equal strength of the two choice representations to a state, in which one strongly dominates. (At least, that is how we understand the question of the reviewer.) We thank the reviewer for bringing up this question, because it forced us to be clearer in the revised manuscript about this aspect of the decision process (subsection “Action value map in SEF reflects competition between available saccade choices”). On choice trials, there is a simultaneous onset of activity in the two areas of the action value map that correspond with the location of the two target options (Figure 6). Throughout the task, there is a robust representation of both options maintained in SEF, even after the divergence of activity that indicates the chosen option and action (Figure 4; Figure 6). However, during the initial rise in activity the SEF population does not indicate the value of the target in its PD. At the time, when the neurons start to differentiate their activity according to PD value they also reflect the value of the NPD value. This can be seen very clearly in Figure 4 showing the activity of SEF neurons for PD targets of medium value. Depending on the value of the NPD target, the activity starts to change ~110-120 ms after target onset. The activity in this figure is not sorted by saccade choice, but it can be presumed that the monkey chose the PD target, when the NPD target was smaller in value (yellow line) and chose the NPD target, when it was larger in value (red line). However, it took this much time for the SEF neurons to indicate value even during no-choice trials, when there was no competing target. It seems therefore that the SEF neurons always indicate relative action value. There is therefore no time period in which two populations of SEF neurons represent the absolute action value of a target independent of the value of any competing target. Nevertheless, there is clear evidence of a succession of an initial undifferentiated state to a more and more differentiated state in which the chosen action value representation increases in activity and the non-chosen one decreases (again see Figure 4). Thus, there is clear evidence for a dynamic process as would be expected by a decision mechanism driven by competition via inhibitory interactions.

[Editors' note: further revisions were requested prior to acceptance, as described below.]

*Thank you for resubmitting your work entitled "Sequential selection of economic good and action in medial frontal cortex of macaques during value-based decisions" for further consideration at* eLife*. Your revised article has been favorably evaluated by Timothy Behrens (Senior editor), a Reviewing editor, and two reviewers. The manuscript has been improved but there are some remaining issues that need to be addressed before acceptance, as outlined below:*

*1) Both referees have some remaining issues that we would like you to address carefully. 2) The Reviewing editor is unhappy about your reply concerning action value (your point #2 and the text 'The SEF neurons encode therefore the action value of saccades to the PD target'): of the three types of action value that you mention, only the first type is real action value. The other two types are a combination of action values (your type 2) and straight chosen value (your type 3, which is the opposite of action value, a decision output variable). To avoid confusion in the community, please accept the only valid definition of action value, which comes out of machine learning: the value of an action that is independent of the action being chosen (a decision input variable). To me it looks like you did not find coding of action value of your type 1. Please rephrase your description of the type of value being coded.* The reviewer is correct that ‘action value’ is a term that is often used in the literature with slightly varying meaning and that a consistent use of nomenclature is important for the field. We modified therefore our description of the different types of signals that represent the value associated with particular actions.

“There are a number of subtypes of value signals that are associated with actions, such as saccades […] These ‘chosen action value’ signals represent the output of the decision process.”

We fully agree that we found ‘relative action value’, and not ‘action value’ signals using the terminology used in the paragraph above. We rephrased our description accordingly throughout the manuscript.

Reviewer #1:

*My concerns were addressed for the most part. However there remain a few relatively minor issues. The numeration below refers to the points raised in my first review.*

*1) How many neurons are "task-related"? In the first paragraph of the subsection “Task-related neurons” the authors state 353; in the fifth paragraph they state 362. Also, in the rebuttal, they say that the results reported in the paper, obtained with the smaller population of 128 neurons that pass the BIC criterion, are valid also for the larger population of 353/362 task-related cells. This fact should be reported in the paper (the additional figure could be included as supplementary material).*

We thank the reviewer for pointing this out. The number of “task related” neurons is 353. We apologize for the confusion and have corrected the numbers in the manuscript. We also reported the result for the later population. We added a new Figure4—figure supplement 3 and some descriptions (subsection “SEF neurons reflect the value of both choice options in an opposing way”).

*In the rebuttal the authors explain why they fitted responses with a sigmoid. The two reasons – (1) there are cells with "curved" tuning and (2) there are cells with floor/ceiling effects – seem to me one and the same. In any case, the authors should clarify this point in the Methods, not just with this reviewer.*

We have included in the Methods section a description that explains the reasons for using the sigmoidal function and its relationship with linear functions (subsection “Task-related neurons”).

*3) Pooling forced choice trials with other trials. Here the authors did not really address my point. The question is this: If instead of separating forced choices for other trials the authors had pooled all trials, as was done in many other studies, would the results presented here be affected in any significant way?*

We thank the reviewer for pointing out this interesting point. We apologize for not fully address the reviewer’s point in the previous reply. We added a new figure (Figure 7—figure supplement 1) and some discussion in the paper (subsection “Instantaneous changes in SEF activity state space reflect decision process”) to explain why we did not pooled forced choice trial with other trials. We agree that in many studies, no-choice trials can be considered forced choices in which simply one of the two values is zero and all trials can be pooled. This is a reasonable strategy if neuronal activity in no-choice trials is indeed similar to the one in choice trial. However, there are examples in the literature, where there is an obvious difference. In a human psychophysical experiments that we conducted recently, the reaction time distribution for choice trials with zero alternative value (that is two targets, one of which is associated with no reward) are different from those for no-choice trials (Chen, Mihalas, Niebur and Stuphorn, 2013). Alexandre Pastor-Bernier and Paul Cisek (2011) reported activity differences in the premotor cortex between choice and no-choice trials. In our initial analysis, we therefore separated the choice trials from no-choice trials to avoid overlooking some useful information by combining the two conditions together. It turned out that we did find neuronal activity differences between no-choice and choice trials. In contrast to premotor cortex neurons (Alexandre Pastor-Bernier and Paul Cisek, 2011), SEF neurons did show value tuning in no-choice trial similar to what was shown in choice trials. However, SEF neurons showed higher activity in choice trials than in no-choice trials when the targets in the receptive field (RF) were chosen, and lower activity in choice trials than in no-choice trials when the targets in RF were not chosen (see Figure 4 and Figure 6). Therefore, we could not treat no-choice trials as equivalent to a choice trial in which one of the two values is zero. It is worth pointing out that this is different from the activity pattern observed in LIP (Louie and Glimcher, 2012) that indicate a process of divisive normalization. We will address the difference between no-choice trial and choice trials in detail in a following up manuscript with some descriptive models.

The difference between choice and no-choice trials is also apparent in the state space analysis (see the new Figure 7—figure supplement 1 in the revised manuscript). The trajectories show the population neuronal activity when the down-left targets were chosen (the black trajectories in Figure 7 in the main manuscript). The trajectories are grouped according to both chosen and non-chosen values. The red, orange and blue colors indicate the large (L), medium (M) and small (S) chosen values in choice trials. Solid, dash and dotted lines indicate the small, medium and large non-chosen values. Please note that there are fewer trajectories when the target was less valuable, because it was chosen less often. In comparison, purple, black and green colors indicate the large, medium and small chosen value in no-choice trials. The trajectories for choice trials were influenced by both chosen and non-chosen values. Specifically, given a chosen value (indicating by color), the trajectories were slightly lower along the value axis if the non-chosen value was larger (indicating by line patterns). Accordingly, if no-choice trials simply represent a situation, in which the non-chosen value is 0, the trajectories for no-choice trial should be similar to the ones on choice trials with small non-chosen value. Instead, the trajectories for no-choice trials are very different and reach always a smaller point along both the value and direction axis.

Given the neuronal activity differences, we decided to keep the choice and no-choice trials separate and to show the SEF activity on both types of trials were appropriate.

Reviewer #2:

*It would be useful if the authors provided additional discussion, in the manuscript, on the predictions they believe each of the models make, and how their data is consistent/inconsistent with each of the models. They have provided this information in the replies to the reviewers, but it needs to be in the manuscript. Otherwise, I have no further comments.*

We added a more detailed discussion of the different predictions for each model and whether our data support these predictions (Discussion).